 **eLIFE**

# Translesion polymerase kappa-dependent DNA synthesis underlies replication fork recovery

**Peter Tonzi, Yandong Yin, Chelsea Wei Ting Lee, Eli Rothenberg, Tony T Huang***

Department of Biochemistry & Molecular Pharmacology, New York University School of Medicine, New York, United States

**Abstract** DNA replication stress is often defined by the slowing or stalling of replication fork progression leading to local or global DNA synthesis inhibition. Failure to resolve replication stress in a timely manner contribute toward cell cycle defects, genome instability and human disease; however, the mechanism for fork recovery remains poorly defined. Here, we show that the translesion DNA polymerase (Pol) kappa, a DinB orthologue, has a unique role in both protecting and restarting stalled replication forks under conditions of nucleotide deprivation. Importantly, Pol kappa-mediated DNA synthesis during hydroxyurea (HU)-dependent fork restart is regulated by both the Fanconi Anemia (FA) pathway and PCNA polyubiquitination. Loss of Pol kappa prevents timely rescue of stalled replication forks, leading to replication-associated genomic instability, and a p53-dependent cell cycle defect. Taken together, our results identify a previously unanticipated role for Pol kappa in promoting DNA synthesis and replication stress recovery at sites of stalled forks.

DOI: https://doi.org/10.7554/eLife.41426.001

**\*For correspondence:**
tony.huang@nyumc.org

**Competing interests:** The authors declare that no competing interests exist.

## Introduction

Replication stress in dividing cells often leads to increased stalled forks and under-replication of the DNA, which can ultimately contribute to elevated DNA damage, mutagenesis and genomic instability (*Zeman and Cimprich, 2014*). One of the most common human diseases associated with replication stress is cancer (*Bartek et al., 2012*). The Fanconi Anemia (FA) disease is a chromosomal instability disorder that is caused by biallelic mutations in any of the 21 FA genes; FA is characterized by multiple developmental abnormalities, progressive bone marrow failure and cancer predisposition (*Nalepa and Clapp, 2018*; *Ceccaldi et al., 2016*). The FA pathway, which includes FA gene products representing ubiquitin signaling, homology-directed recombination (HDR) and nucleolytic pathways, plays a central role in the repair of DNA interstrand crosslinks (ICLs) and regulates cellular responses to replication stress (*Kottemann and Smogorzewska, 2013*; *Michl et al., 2016a*). The FA pathway is activated via lysine site-specific monoubiquitination of two of its effector FA proteins, FANCD2 and FANCI (*Garcia-Higuera et al., 2001*; *Sims et al., 2007*; *Smogorzewska et al., 2007*). The molecular details as to how the FA pathway protects against replication failures and how this is related to human disease is still poorly understood. Understanding this fundamental process could represent an attractive therapeutic target against cancers.

The FA pathway is strongly activated by hydroxyurea (HU) (*Taniguchi et al., 2002*), an inhibitor of ribonucleotide reductase (RNR), which unlike ICL-inducing agents (such as mitomycin C or cisplatin), does not directly elicit DNA lesions that require removal, but induces severe replication fork slowing or stalling through the depletion of the cellular deoxynucleotide pool (dNTPs). Additionally, recent studies found both FANCD2 and FANCI to be associated with the replisome in response to replication fork arrest and that the activation of the FA pathway is critical for protecting stalled forks from

nucleolytic degradation of nascent DNA during HU treatment, and for promoting fork restart after HU is removed (*Lossaint et al., 2013*; *Schlacher et al., 2012*; *Chen et al., 2015*). How the FA pathway controls fork restart is not well-understood.

The role of translesion DNA polymerases (TLS Pols) as it pertains to the mammalian replication stress response has remained enigmatic. Stalled replication forks that are stabilized by the ATR pathway can be restarted by these error-prone TLS Pols when the source of the stress itself (including bulky DNA adducts or UV crosslinks) cannot be removed in a timely fashion, as in the case of an unrepaired DNA lesion (*Zeman and Cimprich, 2014*). Specifically, the replication machinery can 'bypass' these physical barriers at stalled forks by swapping the replicative Pols for TLS Pols; this is known as the DNA damage tolerance (DDT) pathway that is typically initiated by proliferating cell nuclear antigen (PCNA) monoubiquitination and engagement of a specific Y-family TLS Pol (such as Pol eta, kappa, iota, or Rev1) that possess ubiquitin-binding domains (UBDs) (*Kannouche et al., 2004*; *Bienko et al., 2005*). The DDT pathway enables the recruitment of TLS Pols to bypass, or 'tolerate', the DNA lesions during or post S-phase DNA synthesis (in the absence of repair) in order to complete genome duplication in a timely manner and promote cell survival, but at a cost of increased mutagenesis (*Sale et al., 2012*). Here, we set out to identify factors that regulate fork restart under conditions of nucleotide deprivation. Surprisingly, we found that the TLS Pol kappa (PolK) plays a critical role in fork restart during high-dose HU (2 mM) treatment. Importantly, both the FA pathway and PCNA polyubiquitination are critical regulators of PolK recruitment and function at stalled forks. These findings reveal a previously unappreciated role for a Y-family TLS Pol in promoting replication fork recovery during conditions of nucleotide deprivation.

## Results

### TLS polymerase kappa (PolK) is required for efficient replication fork restart

To understand the molecular basis of individual DNA synthesis events under replication stress conditions, we employed the single-molecule DNA fiber technique to study the role of TLS Pols in modulating efficient fork restart from HU-induced stalled replication forks (*Figure 1A*). Asynchronous-growing RPE-1 or other human cell lines were initially pulse-labeled with iododeoxyuridine (IdU) to mark all elongating replication forks, followed by a wash step and a high-dose HU (2 mM) treatment to cause dramatic slowing or stalling of all replication forks. The HU was then subsequently washed out and replenished with fresh media to initiate replication fork restart in the presence of chlorodeoxyuridine (CldU) (see DNA fiber labeling schematics, *Figure 1A*). The efficiency of fork restart events was quantified by scoring the number of stalled forks (green-only tracks) as compared to restarted forks (green tracks that are immediately followed by red tracks) and calculated as a percentage of stalled forks from the total fork events. Unexpectedly, only PolK, but not Polζ, or other members of the Y-family TLS Pols, is required to promote efficient fork restart after HU treatment, as shown by the increased percentage of stalled forks (*Figure 1A* and *Figure 1—figure supplement 1A*). Based on previous work by Helleday and colleagues (*Petermann et al., 2010*), fork restart efficiency as measured by DNA fiber analysis could be highly dependent on the duration of fork-stalling events (time of HU treatment). This appears to be the case as the ability of PolK to rescue stalled forks is linked to a shorter duration of HU treatment (within 4 hr) (*Figure 1B* and *Figure 1—figure supplement 1B,C*). In contrast, if replication forks are stalled for too long (8 hr or greater), the number of stalled forks increases dramatically even in control cells and they become less dependent on a PolK for fork restart. Under this scenario, stalled forks are more likely to be rescued by dormant origin firing (*Ge et al., 2007*) (data not shown) or by HDR-related factors (*Petermann et al., 2010*).

Next, we wanted to determine whether the PolK-mediated fork restart is specific for rescuing stalled forks caused by nucleotide deprivation. To rule out other potential HU-mediated fork-stalling effects, such as through oxidative stress (*Huang et al., 2016*), we asked whether supplementing the media with deoxynucleosides (dNs) (*Aird et al., 2015*) in the presence of HU (no wash step) could rescue HU-mediated stalled forks as measured by CldU labeling of restarted forks (see schematics, *Figure 1C*). We found that the percentage of stalled forks were very similar when comparing samples with normal fork restart after HU wash off *versus* samples with added dNs under continuous HU treatment (*Figure 1C*). This suggests that replenishing the depleted dNTP pool caused by the high-

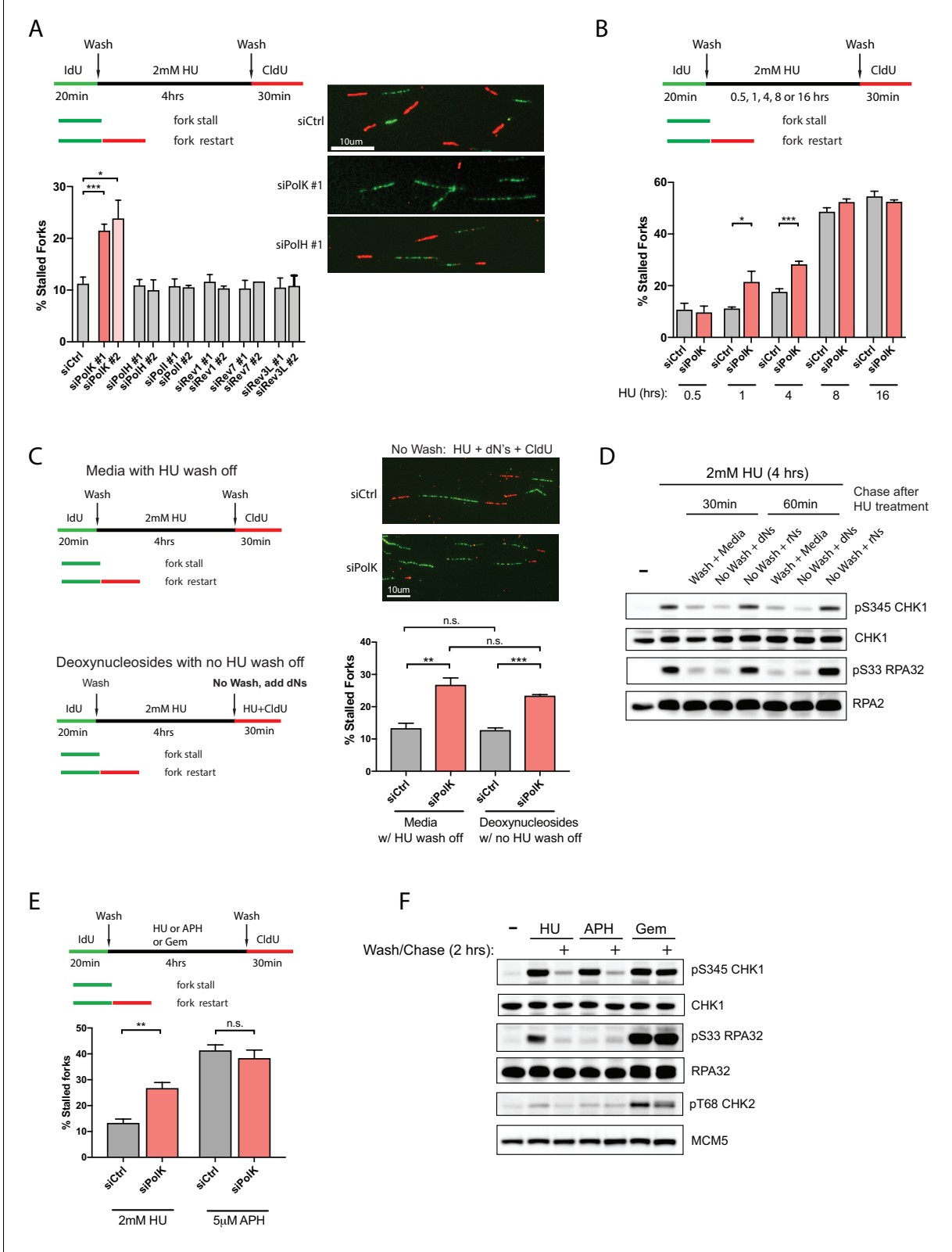

**Figure 1.** PolK is required for replication fork restart due to nucleotide deprivation. (**A**) Schematic for measuring replication fork restart by DNA fiber analysis. Quantification of fork restart efficiency (% stalled forks) in HU-treated (2 mM) RPE-1 cells using two independent siRNAs against individual TLS Pols as indicated. Representative images of the DNA fiber tracts are shown. (**B**) Quantification of fork restart efficiency in RPE-1 cells comparing different HU (2 mM) treatment time-points in the presence or absence of PolK siRNA knockdown. (**C**) Quantification of fork restart efficiency in HU-

*Figure 1 continued on next page*

Figure 1 continued

treated RPE-1 cells with either a wash step with fresh media or with no wash (HU still present) supplemented with 250 µM deoxynucleosides (dNs) for recovery. (D) Western blot analysis of RPE-1 cells treated with 2 mM HU for 4 hr followed by either a wash step with fresh media or no wash (HU still present) supplemented with 250 µM deoxynucleosides (dNs) or 250 µM ribonucleosides (rNs) for 30 or 60 min chase. (E) Quantification of fork restart efficiency comparing fork-stalling agents, HU (2 mM) or APH (5 µM), in the presence or absence of PolK siRNA knockdown. (F) Western blot analysis of RPE-1 cells treated with either HU (2 mM), APH (5 µM), or Gemcitabine (Gem, 1 µM) for 4 hr, followed by a wash step and recovery in fresh media for 2 hr. Data for % stalled forks are represented by mean ± s.d. of three independent experiments and p-values calculated using t-test with Welch's correction. n.s. = no significance, * = $p < 0.05$, ** = $p < 0.01$, *** = $p < 0.001$, **** = $p < 0.0001$.

DOI: https://doi.org/10.7554/eLife.41426.002

The following figure supplements are available for figure 1:

**Figure supplement 1.** siRNA knockdown efficiencies and complementation of CRISPR 293T sgPolK clonal cells.

DOI: https://doi.org/10.7554/eLife.41426.003

**Figure supplement 2.** Gemcitabine-induced stalled forks are not amenable for fork restart assays.

DOI: https://doi.org/10.7554/eLife.41426.004

dose HU treatment for 4 hr with dNs in the media can rescue stalled forks back to the same level as washing out HU. As expected, depletion of PolK elevated stalled forks levels in samples with or without HU wash off supplemented with dNs (*Figure 1C*). The rescue of stalled forks in the presence of high-dose HU correlated with checkpoint recovery (decrease in phosphorylated Ser345 Chk1 and Ser33 RPA32 signals) after supplementing with dNs, but not with ribonucleosides (rNs) (*Figure 1D*). This is consistent with the fact that HU primarily acts as a potent inhibitor of RNR, which prevents the conversion of ribonucleotides to deoxyribonucleotides, leading to nucleotide deprivation and fork stalling. Thus, supplementing with rNs in HU-treated cells will not replenish the dNTP pool and the forks will remain stalled under HU, leading to prolonged checkpoint activation.

Next, we compared whether other fork-stalling agents, such as aphidicolin (APH) or Gemcitabine (Gem), can behave similarly to HU treatment for fork restart after wash off using the same fork restart DNA fiber assay (see schematics, *Figure 1E*). APH is a reversible, potent and specific inhibitor of B-family DNA polymerases (*Vesela et al., 2017*), which includes Polα and the replicative DNA polymerasesδ and ε. Interestingly, in control samples, APH treatment followed by a wash off resulted in higher levels of fork-stalling in comparison to HU treatment, and the depletion of PolK did not further increase fork-stalling events (*Figure 1E*). We speculate that the stalled fork structures in HU-*versus* APH-treated cells are likely processed differently (*Vesela et al., 2017*; *Barlow et al., 2013*) due to the fact that unlike HU, APH treatment does not lead to RPA phosphorylation even though both can activate Chk1 (*Figure 1F*). Gem, on the other hand, acts as a nucleoside analog that blocks DNA synthesis (*Mini et al., 2006*). Under our conditions, we failed to detect fork restart and checkpoint recovery after washing off Gem at various doses, thus precluding any analysis of fork restart (*Figure 1—figure supplement 1A,B*).

## The FA pathway is required for PolK-mediated fork restart

To determine whether PolK functions in the same pathway or in parallel with the FA pathway for fork restart, we used siRNA knockdown strategies in combination with FA patient-derived cells or CRISPR-Cas9-mediated disruption of *PolK* alleles in 293 T cells (sgPolK) to assess the functional link between PolK and the FA pathway. In an extension of our previous findings (*Chen et al., 2015*), FA fibroblasts from FANCD2-deficient patient cells (PD20) showed defective fork restart that could be corrected by FANCD2 WT complementation, but not its monoubiquitination-defective mutant K561R (*Garcia-Higuera et al., 2001*) (*Figure 2A*). However, the additional knockdown of PolK in PD20 vector control or K561R mutant-expressing cells did not further increase the level of stalled forks, suggesting that PolK is likely epistatic to the FA pathway to facilitate fork restart (*Figure 2A*). A Chk1 inhibitor (Chk1i) treatment was utilized as a positive control for replication stress to establish the upper limits of detectable stalled forks in our assay (*Figure 2A*). Importantly, the analysis of one of the 293T *PolK* CRISPR clones (sgPolK #1) demonstrated that GFP-tagged PolK wild-type (WT) expression can rescue defective fork restart, but is incapable of rescue when FANCD2 is simultaneously depleted by siRNA (*Figure 2B*). The ability of PolK to promote fork restart also strongly correlated with longer track lengths of DNA synthesis after HU wash off (quantifying the length of the

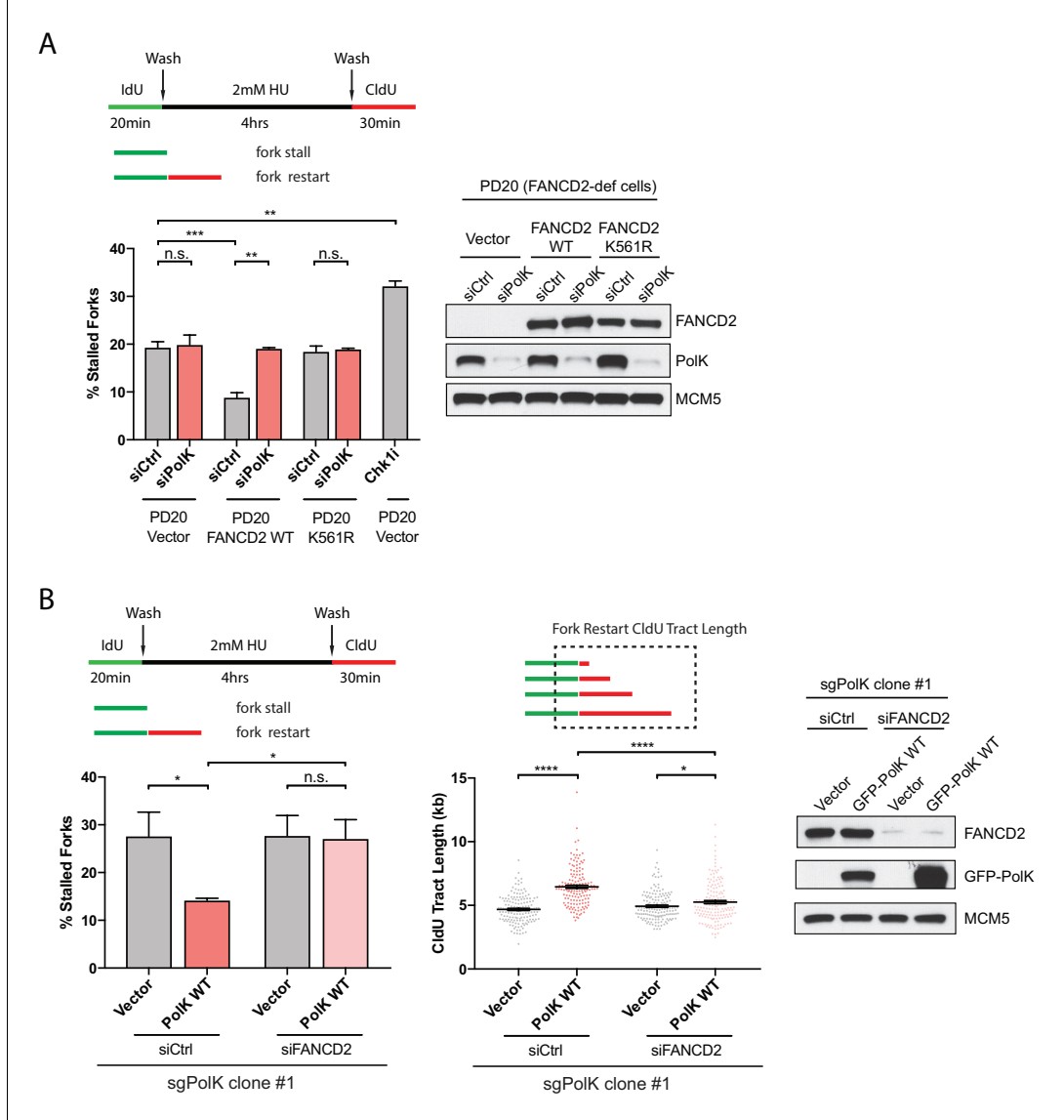

**Figure 2.** PolK functions in concert with the FA pathway to promote replication fork restart. (**A**) Quantification of fork restart efficiency in FANCD2-deficient patient cells (PD20) complemented with either vector only, FANCD2 WT, or K561R mutant in the presence or absence of PolK siRNA and treated as indicated. PD20 (vector only) cells were treated with 300 nM Chk1i (AZD7762) throughout the duration of HU and CldU time points as a positive control for the detection of elevated fork-stalling events. Western blot analysis showing siRNA knockdown efficiency in PD20 cells. (**B**) Quantification of fork restart efficiency in 293T CRISPR PolK (sgPolK) cells complemented with either empty vector or GFP-PolK WT in the presence or absence of FANCD2 siRNA and treated as indicated. CldU (red) tract length measurements of restarted forks determine the varying degree of individual fork restart events. Western blot analysis showing expression and siRNA knockdown efficiency in sgPolK 293 T cells. Data for % stalled forks are represented by mean ± s.d. of three independent experiments and p-values calculated using t-test with Welch's correction. Data for tract length measurements are plotted from three independent experiments with mean ± s.e.m. and p-values calculated using Mann-Whitney t-test. n.s. = no significance, * = p < 0.05, ** = p < 0.01, *** = p < 0.001, **** = p < 0.0001.
DOI: https://doi.org/10.7554/eLife.41426.005

red tracks only in fork events containing both green and red tracks) (*Figure 2B*). Taken together, these data suggest that PolK-mediated fork restart requires the activation of the FA pathway.

## Loss of PolK does not affect replication checkpoint response or recovery

Next, we wanted to determine whether the failure to properly restart forks in PolK-deficient cells would somehow impact the replication checkpoint response or recovery by monitoring both Chk1 and RPA phosphorylation levels in response to HU treatment. A previous study showed that PolK was important for facilitating Chk1 phosphorylation in response to replication stress (*Bétous et al., 2013*). Surprisingly, using our experimental setting, both ATR-dependent Chk1 phosphorylation on Ser345 or RPA32 phosphorylation on Ser33 (or Ser4/8) were unaffected by the loss of PolK or FANCD2 (*Figure 3A* and *Figure 3—figure supplement 1A*) in different cell lines. Even in the 293T *PolK* CRISPR clone (sgPolK clone #1), the reconstitution of GFP-PolK WT did not alter the kinetics of checkpoint recovery of phosphorylated Chk1 or RPA after HU wash off (*Figure 3B*). However, the levels of PCNA monoubiquitination (mUb-PCNA) was higher in the GFP-PolK WT-expressing sgPolK clone #1 293 T cells in comparison to the vector control cells (*Figure 3B*). As PolK has been previously shown to directly interact with mUb-PCNA (*Jones et al., 2012*; *Guo et al., 2008*), this suggests that expression of PolK may help stabilize mUb-PCNA in response to HU treatment.

## PolK interacts with K48-linked polyubiquitinated PCNA in HU-treated cells

To determine whether PolK interacts with mUb-PCNA in response to HU treatment, we used a formaldehyde-crosslinking and immunoprecipitation (IP) technique (*Kannouche et al., 2004*) to capture transiently associated proteins on PolK. We found that GFP-tagged PolK WT, but not mutations of both UBDs (UBZ mutant), was able to interact with mUb-PCNA in an HU-dependent manner (*Figure 3C*). However, the interaction of PolK with FANCD2, FANCI, RPA, or even unmodified PCNA, was not HU-inducible, nor was it dependent on its UBDs (*Figure 3—figure supplement 1B*). A commercially available antibody against mUb-PCNA was used both for western blot analysis and crosslink-IP to verify that PolK interacts with mUb-PCNA in a manner that is dependent on its UBDs (*Figure 3C* and *Figure 3—figure supplement 1C*). The detection of additional higher molecular weight mUb-PCNA antibody-reactive bands in the anti-GFP antibody pull-down assay suggests that PolK is likely interacting with differentially polyubiquitinated (di- or tri-ubiquitinated) forms of PCNA (*Figure 3C*). Although several studies suggest that DNA damage-induced polyubiquitination of PCNA is primarily composed of K63-linked Ub chains (*Lin et al., 2011*; *Ciccia et al., 2012*), it is unclear whether HU-induced polyubiquitinated PCNA follows the same rule. To characterize the nature of the ubiquitin chain linkage on PCNA, polyubiquitinated PCNA was enriched and purified from HU-treated cells using either the anti-GFP or mUb-PCNA antibody pulldown assay (*Figure 3D* and *Figure 3—figure supplement 2B*) and the isolates were subjected to an in vitro ubiquitin chain restriction analysis (*Mevissen et al., 2013*). Surprisingly, when using a purified SARS-PLpro deubiquitinating (DUB) enzyme that exclusively cleaves longer K48-linked ubiquitin chains (but not monoubiquitinated substrates) (*Békés et al., 2016*; *Békés et al., 2015*), the higher molecular weight species of PCNA (likely di- or triUb-PCNA) were greatly reduced when incubated with PLpro WT, but not its catalytic mutant (C112A) or the AMSH K63 chain cleavage-specific metalloprotease (*Figure 3D* and *Figure 3—figure supplement 2B*). For positive control, the catalytic domain of the ubiquitin protease USP2 was able to non-specifically cleave both the monoUb- and diUb-PCNA bands in the assay (*Figure 3D*). The ubiquitin chain linkage specificity of the different recombinant DUBs tested here was reconfirmed using an in vitro cleavage assay of either K48 or K63 tri-Ub unanchored chains as substrates (*Figure 3—figure supplement 2A*). Using this in vitro ubiquitin chain restriction analysis, it supports the notion that PolK mostly interacts with K48-linked polyubiquitinated PCNA in HU-treated cells.

Since the UBDs of PolK is required for the recruitment of PolK to polyubiquitinated PCNA in response to HU treatment, we next asked whether the UBDs of PolK is required for proper fork restart. Using the CRISPR sgPolK cells that are transiently transfected with either GFP-tagged PolK WT or the UBZ mutant, we showed that the ubiquitin-binding capacity of PolK is required for PolK-mediated fork restart (*Figure 3E*). Importantly, the catalytically dead (CD) mutant of PolK (DE198/199AA) was also unable to properly perform fork restart. This demonstrates that both the DNA polymerase activity and ubiquitin-mediated transactions (likely through polyubiquitinated PCNA) are essential for PolK-dependent recovery of stalled replication forks (*Figure 3E*). In line with the

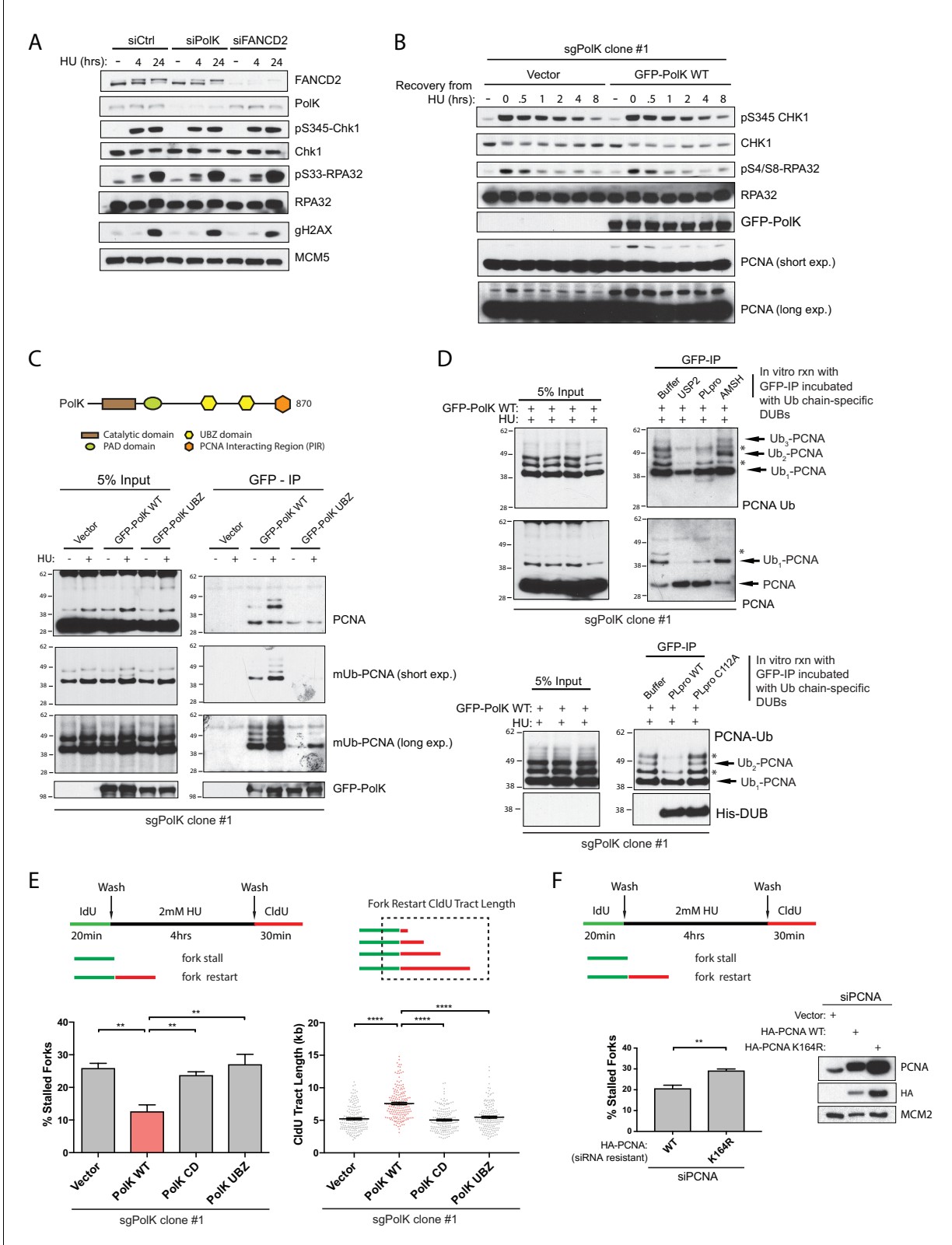

**Figure 3.** PolK interacts with K48-linked polyubiquitinated PCNA via its UBZ domains in a HU-dependent manner. (**A**) Western blot analysis of RPE-1 cells treated with the indicated siRNAs and HU (2 mM) time-points. (**B**) Western blot analysis of 293T sgPolK cells complemented with either empty vector or GFP-PolK WT and pulsed with HU (2 mM) for 4 hr before wash step and recovery for the indicated time-points. (**C**) Schematic diagram showing domains of PolK. Formaldehyde-induced crosslinking of 293T sgPolK cells treated with HU (2 mM) for 4 hr as indicated. Extracts from cells

*Figure 3 continued on next page*

*Figure 3 continued*

complemented with either empty vector, GFP-PolK WT or a double ubiquitin-binding domain mutant (UBZ) of GFP-PolK were then subjected to anti-GFP pulldown, followed by Western blot analysis with the indicated antibodies. (**D**) Ubiquitin chain restriction digest analysis using similarly treated and immunoprecipitated (IP) samples as in (**C**) to enrich for polyubiquitinated PCNA that is bound by GFP-PolK and induced by HU. Samples on beads were then incubated with 900 ng of indicated recombinant DUBs for 1 hr at 37°C prior to Western blot analysis with the indicated antibodies (upper and lower panels). SARS PLpro catalytic mutant (C112A) was used for negative control as indicated (lower panel). (**E**) Quantification of fork restart efficiency in 293T sgPolK cells complemented with either empty vector, GFP-PolK WT, Catalytic-Dead (CD), or ubiquitin-binding mutant (UBZ). CldU (red) tract length measurements of restarted forks were determined for WT and the different PolK mutants. (**F**) Quantification of fork restart efficiency in U2OS cells treated with PCNA siRNA and complemented with siRNA-resistant HA-tagged PCNA-WT or ubiquitin site mutant HA-PCNA K164R. Western blot analysis showing exogenously expressed siRNA-resistant HA-PCNA in U2OS cells. Data for % stalled forks are represented by mean ± s.d. of three independent experiments and p-values calculated using t-test with Welch's correction. Data for tract measurements are plotted from three independent experiments with mean ± s.e.m. and p-values calculated using Mann-Whitney t-test. n.s. = no significance, * = p < 0.05, ** = p < 0.01, *** = p < 0.001, **** = p < 0.0001.

DOI: https://doi.org/10.7554/eLife.41426.006

The following figure supplements are available for figure 3:

**Figure supplement 1.** PolK interacts with FANCD2 and RPA independently of its UBZ domain.

DOI: https://doi.org/10.7554/eLife.41426.007

**Figure supplement 2.** HU-dependent PCNA polyubiquitination is susceptible to an in vitro K48-specific polyUb DUB cleavage reaction.

DOI: https://doi.org/10.7554/eLife.41426.008

requirement of the UBDs of PolK for proper fork restart, we also showed that the ubiquitination mutant (K164R) of PCNA has compromised fork restart after HU treatment (*Figure 3F*).

## PolK protects forks against SMARCAL1- and MRE11-dependent nascent DNA degradation

Replication forks that stall after encountering DNA lesions or during conditions of high dose HU treatment may undergo an intermediate fork reversal step to help promote fork protection (*Zellweger et al., 2015*; *Vujanovic et al., 2017*). The regressed arm of a reversed fork resembles a one-ended double-strand break (DSB) and must be properly protected against nucleolytic degradation of nascent DNA. Previous studies demonstrated the requirement of the FA pathway and other HDR proteins, such as Rad51 and the breast cancer susceptibility proteins BRCA1 and BRCA2, for the protection of stalled forks against nascent DNA degradation (*Schlacher et al., 2012*; *Schlacher et al., 2011*). Whether PolK plays a role in fork protection is unknown. To directly measure the extent of nascent DNA degradation of a stalled fork by DNA fiber analysis, we modified the labeling procedure whereby IdU and CldU are first sequentially pulse-labeled, followed by washes and treatment with high dose HU (2 mM) for 4 hr; the red 2nd color track lengths (CldU) were measured to determine the level of nascent DNA degradation (see schematics in *Figure 4A*). In agreement with previous findings (*Schlacher et al., 2012*; *Schlacher et al., 2011*; *Ray Chaudhuri et al., 2016*), we showed that FANCD2 and other HDR-related factors, Rad51 and BRCA2, are required for fork protection (*Figure 4A* and *Figure 4—figure supplement 1A,C and D*). However, in contrast to FANCD2, both Rad51 and BRCA2 are not required for efficient fork restart (*Figure 4—figure supplement 1B*). This suggests that mechanisms involving HDR to prevent fork collapse, such as fork reversals, are separable from fork restart as defined by our HU treatment conditions.

As both PolK and FANCD2 function in the same pathway to mediate fork restart, we next tested whether PolK could also have a role in fork protection. Similar to a FANCD2 loss, we found that PolK depletion caused a reduction in the nascent DNA tract length (*Figure 4A* and *Figure 4—figure supplement 1D*). Additionally, expression of GFP-PolK WT back into the CRISPR sgPolK cells can reverse nascent DNA degradation in HU-treated cells (*Figure 4B*). Intriguingly, the expression of the CD mutant, but not the UBZ mutant of PolK, can still partially rescue PolK deficiency in preventing nascent DNA degradation (*Figure 4B*). This may be due to the fact that the inactive form of PolK is still present in the cells and may bind with high affinity to the nascent DNA, thereby blocking access of nucleases (or other fork remodeling proteins) to the nascent DNA for degradation. This argues that the polymerase activity is not as critical for preventing nascent DNA degradation as it is for mediating fork restart efficiency, thus revealing a possible mechanistic difference between PolK's role in fork restart *versus* fork protection. It is noted that the expression of the UBZ mutant has a

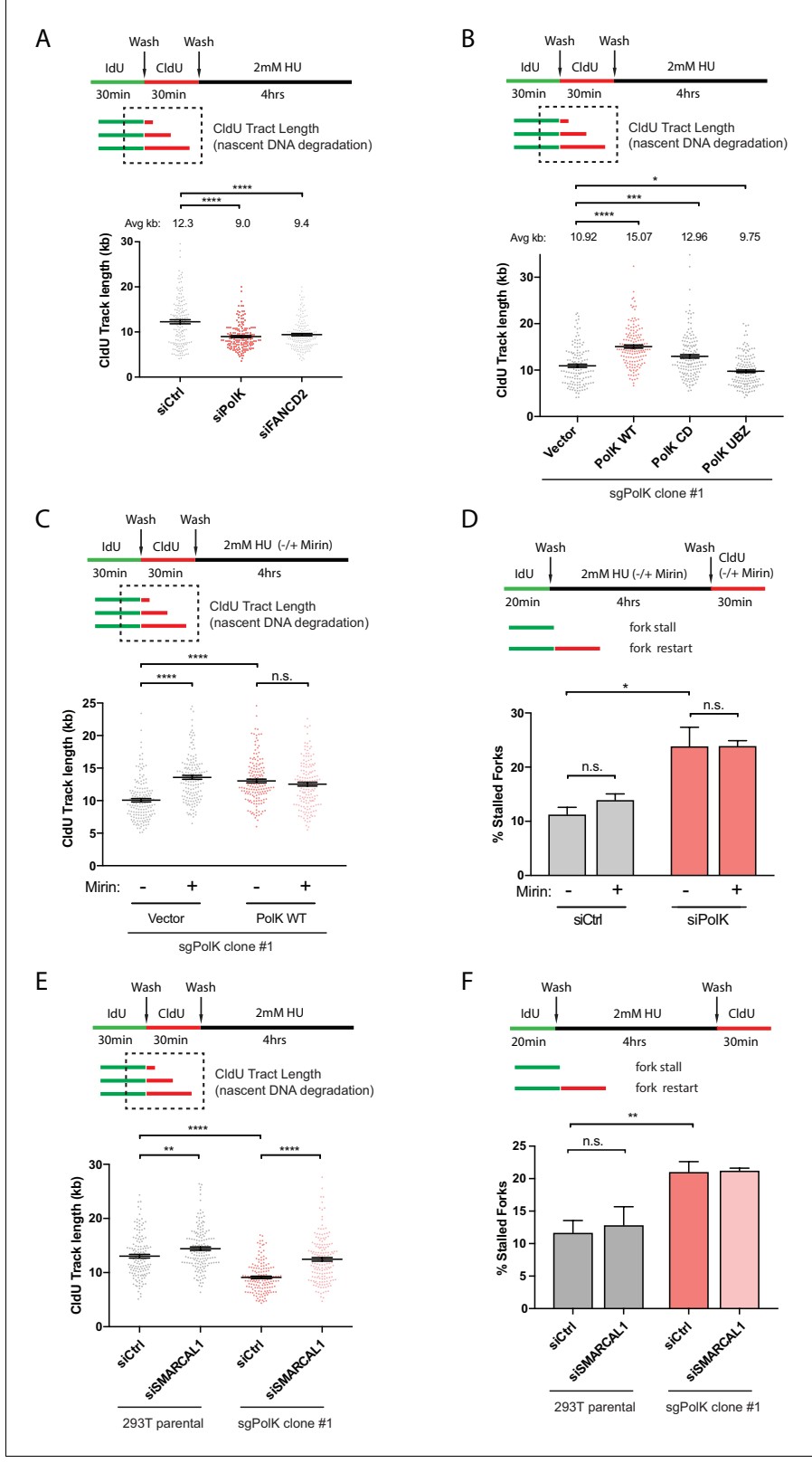

**Figure 4.** PolK prevents MRE11-dependent nascent DNA degradation. (**A**) Schematic for measuring nascent DNA degradation (shortened CldU-labeled tracts) by DNA fiber analysis (**A–C,E**). Quantification of nascent DNA degradation (changes in CldU tract lengths) in 293 T cells treated with the indicated siRNAs. (**B**) Quantification of nascent DNA degradation in 293T sgPolK cells that were complemented with either empty vector, GFP-PolK WT

*Figure 4 continued on next page*

*Figure 4 continued*

or the indicated GFP-PolK mutants. (**C**) Quantification of nascent DNA degradation in 293T sgPolK cells complemented with either empty vector or GFP-PolK WT were treated with or without Mre11 inhibitor, Mirin (50 μm), in the presence of HU as indicated. (**D**) Quantification of fork restart efficiency in RPE-1 cells with the indicated siRNAs and treated with or without Mirin (50 μM) in the presence of HU (2 mM) as indicated. (**E**) Quantification of nascent DNA degradation in parental 293T or 293T sgPolK cells treated with the indicated siRNAs. (**F**) Quantification of fork restart efficiency in parental 293T or 293T sgPolK cells treated with the indicated siRNAs. Data for % stalled forks are represented by mean ± s.d. of three independent experiments and p-values calculated using t-test with Welch's correction. Data for tract length measurements are plotted from three independent experiments with mean ± s.e.m. and p-values calculated using Mann-Whitney t-test. n.s. = no significance, * = p < 0.05, ** = p < 0.01, *** = p < 0.001, **** = p < 0.0001.

DOI: https://doi.org/10.7554/eLife.41426.009
The following figure supplement is available for figure 4:

**Figure supplement 1.** RAD51 and BRCA2 depletion in RPE-1 cells has no effect on replication fork restart after HU treatment.
DOI: https://doi.org/10.7554/eLife.41426.010

marginally more severe phenotype than vector control, implying that the UBZ mutant may still interact with PCNA or some other factor(s) that could lead to increased nascent DNA degradation.

The MRE11 nuclease was previously shown to be critical for nascent DNA degradation of stalled replication forks in FA pathway- or BRCA2-deficient cells (*Schlacher et al., 2012*; *Schlacher et al., 2011*). Recent data also suggests that the SNF2-family of fork remodelers, such as SMARCAL1, acts upstream of MRE11 and is responsible for the processing of stalled forks to the reversed fork structure, enabling MRE11-dependent nucleolytic degradation of nascent DNA (*Taglialatela et al., 2017*). To test whether MRE11 or SMARCAL1 is responsible for mediating nascent DNA degradation in PolK-deficient cells, we treated CRISPR sgPolK cells with either mirin to inhibit MRE11 nuclease activity or with an siRNA against SMARCAL1 during HU treatment. Both mirin treatment and SMARCAL1 knockdown reversed the nascent DNA degradation observed in HU-treated PolK-deficient cells (*Figure 4C,E*). Since we showed that PolK perform dual functions in both fork protection and fork restart, it is unknown whether these two events are functionally connected. For example, would the rescue of fork protection in PolK-deficient cells with either mirin treatment or SMARCAL1 depletion alter the efficiency of fork restart? Remarkably, inhibition of either MRE11 activity or SMARCAL1 had no effect on fork restart after HU treatment (*Figure 4D,F*). This implies that PolK is capable of rescuing stalled forks via two non-overlapping mechanisms: 1) fork protection against SMARCAL1- and MRE11-dependent nucleolytic degradation, and 2) replication fork restart.

## PolK-mediated DNA synthesis during conditions of nucleotide starvation

Past studies have suggested that PolK can cause a reduction in replication fork speed when it becomes aberrantly recruited to the replication fork by either the loss of USP1 or p21$^{CDKN1A}$ in untreated cells (*Jones et al., 2012*; *Mansilla et al., 2016*). Indeed, by DNA fiber analysis, we show that the tract length is slightly elevated (faster fork speed) in the absence of PolK, while USP1 knockdown can lead to a PolK-dependent slow-down of the replication fork in the absence of HU treatment, as demonstrated in our previous study (*Jones et al., 2012*) (*Figure 5—figure supplement 1A*). Thus, in unperturbed, dividing cells, the aberrant recruitment of PolK to the replication fork can lead to reduced DNA synthesis and genomic instability (*Jones et al., 2012*; *Mansilla et al., 2016*), which is likely due, in part, to the slowing of the normal fork speed. However, it is unclear whether PolK plays an important role in DNA synthesis during conditions of nucleotide deprivation. Cells treated with high-dose HU (2 mM or higher) dramatically reduces, but does not completely abolish, DNA synthesis (*Dungrawala et al., 2015*). For instance, in bacteria, Y-family DNA Pols have the potential to operate efficiently at low dNTP concentrations in comparison to the replicative DNA Pols due to intrinsic differences in $K_m$ values for dNTPs (*Godoy et al., 2006*). The slower fork movement in HU-treated cells can be monitored by DNA fiber analysis using a longer labeling time for CldU (see labeling schematics in *Figure 5A*). We found that PolK depletion further reduced DNA

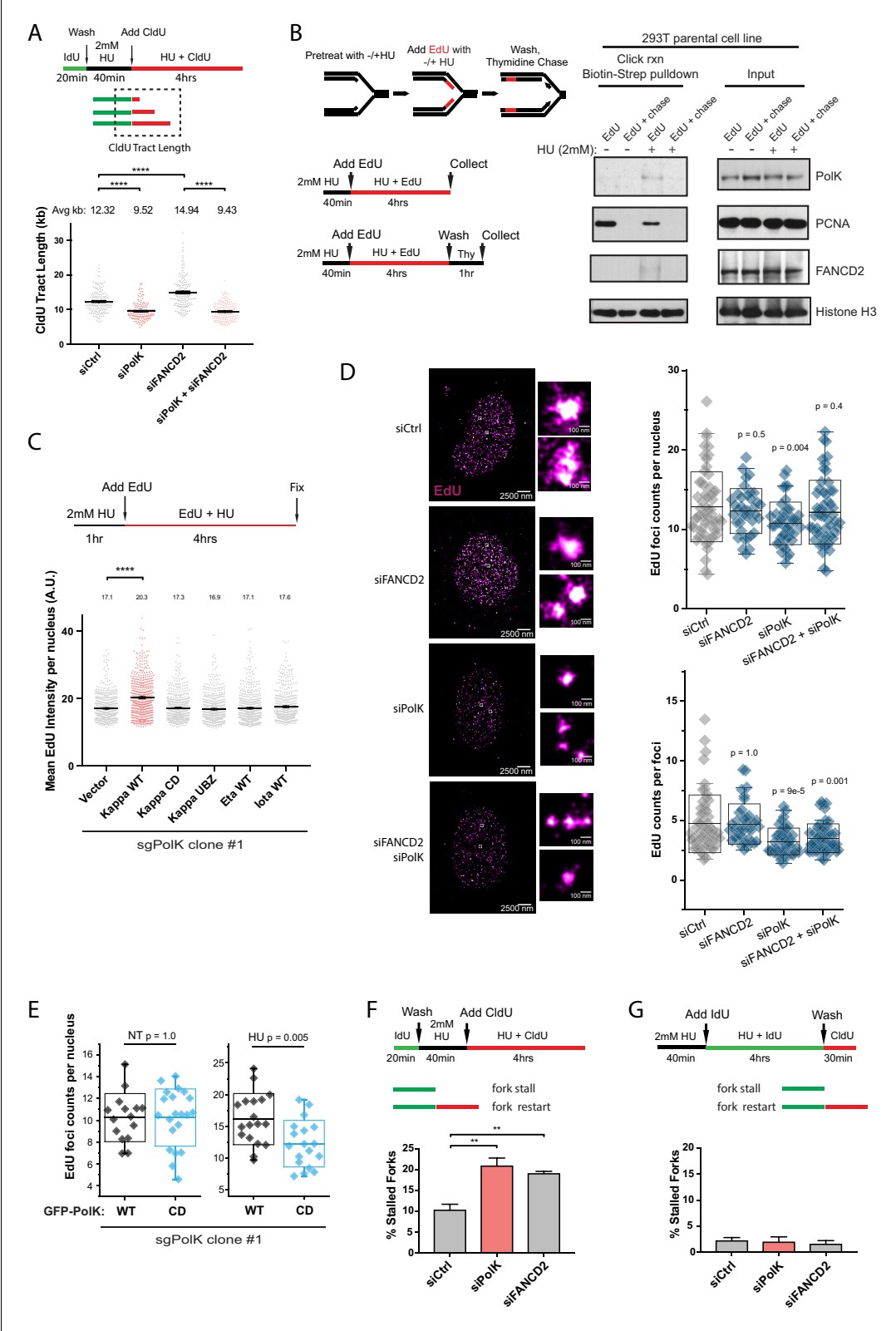

**Figure 5.** PolK-dependent DNA synthesis in the presence of HU. (**A**) Schematic for measuring replication fork speed (CldU tract length) under HU treatment for 4 hr. To ensure that the CldU-labeled DNA was under constant high dose HU treatment, cells were pre-treated with 2 mM HU for 40 min prior to the addition of CldU in the presence of HU for 4 hr. Quantification of fork speed in RPE-1 cells treated with the indicated siRNAs. (**B**) Schematic of a modified iPOND assay to measure proteins associated with the replisome under nucleotide starvation conditions. Similar to (**A**), 293 T cells were

*Figure 5 continued on next page*

*Figure 5 continued*

either left untreated or pretreated with HU (2 mM) for 40 min prior to the addition of EdU (10 µM) in the presence of HU for 4 hr. Cells were then either collected immediately (EdU samples) or chased with Thymidine (10 µM) for 1 hr (EdU + chase) in the presence or absence of HU. Samples without HU were treated for only 10 min with EdU or chased with Thymidine for 1 hr. Western blot analysis showing the biotin-streptavidin pulldown after click-reaction in parental 293 T cells and probed with the indicated antibodies. (C) Schematic for measuring EdU incorporation intensity by direct fluorescence measurements in HU-treated cells. U2OS sgPolK cells were complemented with either empty vector, GFP-PolK WT, GFP-PolK mutant constructs, or different Y-family TLS Pols, GFP-Pol eta or GFP-Pol iota. Cells were pretreated with HU (2 mM) for 1 hr, prior to the addition of EdU (10 µM) in the presence of HU for 4 hr. Mean EdU intensity per nucleus measured by ImageJ were plotted from three independent experiments. (D) Single-molecule localization imaging of EdU signal distribution per foci or nuclei. RPE-1 cells were treated with the indicated siRNAs prior to pretreatment with HU (2 mM) for 1 hr, followed by the addition of EdU in the presence of HU for 4 hr. Representative super-resolution images of nuclei with EdU signal in magenta are shown. Quantification of EdU foci counts per nuclei and amount of EdU counts per foci are plotted from three independent experiments. (E) U2OS sgPolK cells complemented with either GFP-PolK WT or GFP-PolK CD were pulse-labeled with EdU and treated with HU (2 mM) or not (NT). Treatment conditions and quantification of EdU foci per nuclei by super-resolution imaging techniques were done as in (D). (F) Quantification of fork restart efficiency in RPE-1 cells treated with the indicated siRNAs whereby 'restarted' forks are measured as previously elongating forks (IdU tracts) that become converted to CldU tracts in the presence of HU (2 mM). Cells were pretreated with HU for 40 min prior to the addition of CldU for 4 hr to ensure that CldU pulse-labeled cells were already under constant high-dose HU treatment. (G) Quantification of fork restart efficiency in RPE-1 cells treated with the indicated siRNAs whereby IdU pulse-labeled forks under constant HU treatment are measured to determine whether they can be 'restarted' after HU wash off (CldU pulse-label). Data for % stalled forks and quantification of EdU foci counts by single-molecule localization imaging are represented by mean ± s.d. of three independent experiments and p-values calculated using t-test with Welch's correction and indicated above the plots. Data for tract length measurements are plotted from three independent experiments with mean ± s.e.m. and p-values calculated using Mann-Whitney t-test. n.s. = no significance, * = p < 0.05, ** = p < 0.01, *** = p < 0.001, **** = p < 0.0001.

DOI: https://doi.org/10.7554/eLife.41426.011

The following source data and figure supplements are available for figure 5:

**Source data 1.** EdU intensity per nucleus.
DOI: https://doi.org/10.7554/eLife.41426.015
**Figure supplement 1.** Mirin treatment does not affect replication fork speed in either PolK- or FANCD2-depleted RPE-1 cells.
DOI: https://doi.org/10.7554/eLife.41426.012
**Figure supplement 2.** PolK-dependent DNA synthesis under HU is unaffected by DRB treatment.
DOI: https://doi.org/10.7554/eLife.41426.013
**Figure supplement 3.** Single-molecule localization image of EdU signal, PCNA, and GFP-PolK in U2OS cells.
DOI: https://doi.org/10.7554/eLife.41426.014

synthesis (or fork speed) during HU treatment, as measured by reduced CldU tract length adjacent to IdU tracts (*Figure 5A* and *Figure 5—figure supplement 1B*).

The slower fork movement in HU-treated cells is likely not due to unscheduled replication-transcription conflicts as treatment with an inhibitor of the elongation step during RNA Pol II transcription, 5,6-Dichlorobenzimidazole 1-β-D-ribofuranoside (DRB), which is known to reduce replication-transcription conflicts leading to R-loop formation (*Kotsantis et al., 2016*; *Macheret and Halazonetis, 2018*), had no effect on tract length (fork speed) under high-dose HU (2 mM) treatment (*Figure 5—figure supplement 2A*) in control or PolK-depleted samples. Consistent with this result, treatment of RPE-1 cells with either DRB or flavopiridol (FVP) (*Bensaude, 2011*) in the presence or absence of PolK did not cause any changes to HU-mediated checkpoint activation (*Figure 5—figure supplement 2B*).

Previous studies showed that FANCD2 is necessary for actively restraining DNA synthesis upon nucleotide deprivation (*Lossaint et al., 2013*; *Chen et al., 2015*; *Michl et al., 2016b*). In FANCD2-depleted cells, the faster fork speed (longer CldU tracts) in the presence of HU is largely dependent on PolK (*Figure 5A*). Inhibition of MRE11 activity by mirin did not affect the average tract lengths in PolK- or FANCD2-depleted cells (*Figure 5—figure supplement 1C*), suggesting that aberrant nuclease activity acting on deprotected or reversed forks does not account for changes in DNA synthesis levels (fork speed) during nucleotide starvation conditions. One interpretation of these results is that the reduced DNA synthesis during HU treatment in PolK-depleted cells could be due to a defect in fork restart, such that during the time of HU treatment, replication forks will continuously stall and restart and the reduced efficiency of restart associated with the loss of PolK could lead to shorter CldU tracts. While we cannot formally rule out this possibility, we do not favor this model based on our evidence that even though both PolK- and FANCD2-depleted cells have fork restart defects (*Figure 2*), the loss of FANCD2 showed an increase in CldU synthesis during HU treatment

(*Lossaint et al., 2013*; *Chen et al., 2015*), which is the opposite of what has been observed in PolK-deficient cells (*Figure 5A*). This lack of correlation between fork restart and HU-mediated DNA synthesis will be addressed below (*Figure 5F,G*).

Isolation of protein on nascent DNA (iPOND) is a powerful technique to identify proteins bound to nascent DNA in the vicinity of the replisome during unperturbed replication (*Sirbu et al., 2012*). To determine whether PolK directly interacts with the replisome in HU-treated cells, we modified the iPOND protocol by pretreating cells with HU to slow DNA synthesis prior to labeling with the nucleotide analog 5-Ethynyl-2'-deoxyuridine (EdU) in the same media. After click chemistry with biotin, followed by streptavidin beads pull-down, we showed that both PolK and FANCD2 were significantly enriched in the replisome of HU-treated samples, but not present in thymidine-chased samples (*Figure 5B*).

After showing that PolK was directly bound to EdU-labeled nascent DNA, we wanted to determine whether EdU labeling efficiency can be utilized as an additional measurement for DNA synthesis during HU treatment. We pulse-labeled cells with EdU and measured differential EdU fluorescent intensity in U2OS cells (see schematics in *Figure 5C* and *Figure 5—figure supplement 2D*). It is noted that due to the high level of EdU incorporation under normal DNA synthesis, the change in EdU intensity per nuclei could not be observed in PolK-deficient cells unless cells were pretreated with HU prior to the EdU labeling step under HU treatment (*Figure 5—figure supplement 2D*). In CRISPR sgPolK U2OS cells, the expression of PolK WT, but not the CD or UBZ mutant, promoted DNA synthesis during HU treatment (*Figure 5C* and *Figure 5—figure supplement 2C–E*). This was not due to the over-compensation of any TLS Pols as expression of other Y-family TLS Pols, such as Pol eta or iota, could not compensate for the reduced DNA synthesis in PolK-deficient cells under HU treatment (*Figure 5C* and *Figure 5—figure supplement 2E*). Thus, amongst the different TLS Pols in mammalian cells, PolK possesses a unique role in facilitating DNA synthesis under conditions of nucleotide starvation.

Since PolK-dependent DNA synthesis is largely restricted by the presence of FANCD2 in HU-treated cells (*Figure 5A*), we wanted to explore whether the organization of EdU-incorporated nuclear foci is affected by the presence or absence of PolK and/or FANCD2. To quantify the EdU incorporation per nucleus as well as within each replication focus, we employed single-molecule stochastic blinking super-resolution imaging (*Rust et al., 2006*) of fluorescently labeled EdU along with unbiased Pair-Correlation Function analysis (see Materials and methods) (*Sengupta et al., 2011*; *Veatch et al., 2012*). This approach enabled us to map the precise molecular coordinates of EdU molecules within a cell with a resolution of ~9 nm, and extract robust metrics such as the exact amount of EdU foci incorporated per nucleus as well as the amount of EdU content within each focus. Using a labeling scheme to measure HU synthesis (1 hr pretreatment with 2 mM HU, followed by the addition of 10 µM EdU to HU medium), we observed that the total number of foci and the amount of EdU counts per foci are both reduced in PolK-depleted cells (*Figure 5D*). This suggests that PolK has a role in maintaining the total number of active forks per nucleus that is able to synthesize under low nucleotide conditions, in addition to a reduction of synthesis at individual forks. While the number of EdU foci per nucleus is reduced in PolK-depleted cells, this could be rescued when FANCD2 was depleted concomitantly (*Figure 5D*). This suggests that in the absence of PolK, the loss of FANCD2 may enable additional forks (more EdU foci per nucleus) to undergo DNA synthesis in the presence of HU. Thus, FANCD2 may behave as a general restriction factor against aberrant DNA synthesis in response to replication stress. Addressing this point in the future may require DNA locus-specific or genome-wide analysis of fork restart differences within PolK- and/or FANCD2-deficient background. However, the EdU counts per foci (EdU molecule intensity of individual foci) remains decreased in the absence of PolK, irrespective of FANCD2 levels (*Figure 5D*). This implies that while forks that may incorporate EdU independently of PolK, it will still do so at a reduced efficiency (less EdU counts per foci). In the absence of FANCD2, the EdU counts per foci is not elevated above control levels. Since the loss of FANCD2 led to longer tract lengths using DNA fiber assays (*Figure 5A*), this would imply that the EdU counts per foci observed by super-resolution imaging is qualitatively distinct from tract lengths generated by individual forks. What is the exact nature of these EdU counts per foci remains to be determined. Nevertheless, the EdU foci counts per nuclei that is generated by PolK is still dependent on its catalytic activity (*Figure 5E*). Cross-Pair-Correlation analysis between GFP-PolK and EdU signal was done to ensure that differences observed in

EdU foci counts per nucleus were only observed under HU treatment, irrespective of differences in GFP-PolK WT or CD expression levels (*Figure 5E* and *Figure 5—figure supplement 3A*).

Next, we tested whether PolK-dependent DNA synthesis under HU is functionally linked to fork restart. Modifying our DNA fiber analysis (see schematics in *Figure 5F*), we first determined how many of the elongating forks (labeled by IdU, no treatment) remained 'moving' (albeit at a slower velocity) when exposed to HU (pretreated with HU, followed by labeling with CldU for a 4 hr HU treatment). Applying this labeling schematic, the loss of PolK or FANCD2 recapitulated the fork restart defect observed originally when fork restart was measured after HU wash off (*Figure 5F*). Next, we determined how many of the slow-moving forks during HU treatment are converted to restarted forks once HU has been removed. Surprisingly, greater than 95% of the slow-moving forks (pretreated with HU, followed by labeling with IdU for a 4 hr HU treatment) were converted to fast-moving restarted forks (labeled by CldU after HU wash off) irrespective of PolK or FANCD2 deficiency (*Figure 5G*). In summary, the defect in fork restart efficiency in either PolK- or FANCD2-depleted cells occurs at a much earlier point during the transition from normal elongating forks in untreated cells into slower-moving forks under HU treatment (*Figure 5F*, see working model in *Figure 6D*). Even though less forks are able to resume progression (restart) after pulsing with HU in either PolK- or FANCD2-deficient cells, the remaining forks that are competent to undergo DNA synthesis under HU independently of PolK, appears to be fully proficient to resume normal DNA synthesis once HU is removed.

## PolK promotes cell cycle recovery after replication stress

In this study, we have uncovered several unique properties for PolK in alleviating replication stress. Despite significant defects in replication fork protection and fork restart in response to HU treatment, the phenotype in *PolK*-deficient mice in the absence of perturbations is relatively mild (*Burr et al., 2006*; *Stancel et al., 2009*). Asynchronous growing PolK-deficient cells have a normal cell cycle progression in comparison to control cells (*Figure 6—figure supplement 1A*). They also do not experience higher levels of DNA damage or cell cycle checkpoint activation in response to HU and HU wash off (*Figure 3A,B* and *Figure 3—figure supplement 1A*). To reveal a subtler, replication-associated defect, we interrogated how the cells are able to progress and recover from a short-term (4 hr) HU pulse treatment in a time-course study. Using EdU pulse-chase cell cycle analysis to directly visualize the progression of a fluorescently labeled S-phase population into the next cell cycle phase, we observed a prolonged S-phase and a reduced G2 phase in PolK-deficient cells in comparison to control cells (*Figure 6A*). PolK-depleted cells also developed elevated senescence-associated β-galactosidase (SA-βgal) activity after a short-term HU treatment, followed by a 24 hr chase (*Figure 6B*). Importantly, depletion of p53 was capable of reversing both cell cycle delay (data not shown) and SA-βgal activity in PolK-deficient cells (*Figure 6B*). Elevated levels of 53BP1 nuclear bodies in G1 phase cells (Cyclin A-negative) were also observed in PolK-deficient cells after a short-term HU treatment, followed by an 18 hr chase (*Figure 6C*), suggesting that the cause of cell cycle defects is likely due to the gradual accumulation of replication-associated DNA damage that become shielded within 53BP1 nuclear bodies for repair in the next G1 phase (*Lukas et al., 2011*; *Harrigan et al., 2011*). The replication-induced DNA damage phenotype shown in PolK-deficient cells is comparable to those experienced in FANCD2-depleted cells (*Ceccaldi et al., 2012*) (*Figure 6A–C*). Importantly, the siRNA depletion of both Polk and FANCD2 simultaneously yielded similar levels of elevated SA-βgal activity and 53BP1 nuclear bodies as either of the individual siRNA depletions, suggesting that both of these proteins act within the same pathway (*Figure 6C* and *Figure 6—figure supplement 1B*). Significantly, mirin treatment to prevent nascent DNA degradation during the HU pulse treatment did not alter the levels of 53BP1 nuclear bodies (*Figure 6—figure supplement 1C*), suggesting that the replication-associated problems based on the loss of PolK and/or FANCD2 is caused by defects in fork restart efficiency, and not fork protection (MRE11-dependent nascent DNA degradation). Thus, both PolK and the FA pathway cooperate within the same network that resolves replication stress from improperly stalled forks to help maintain genome integrity.

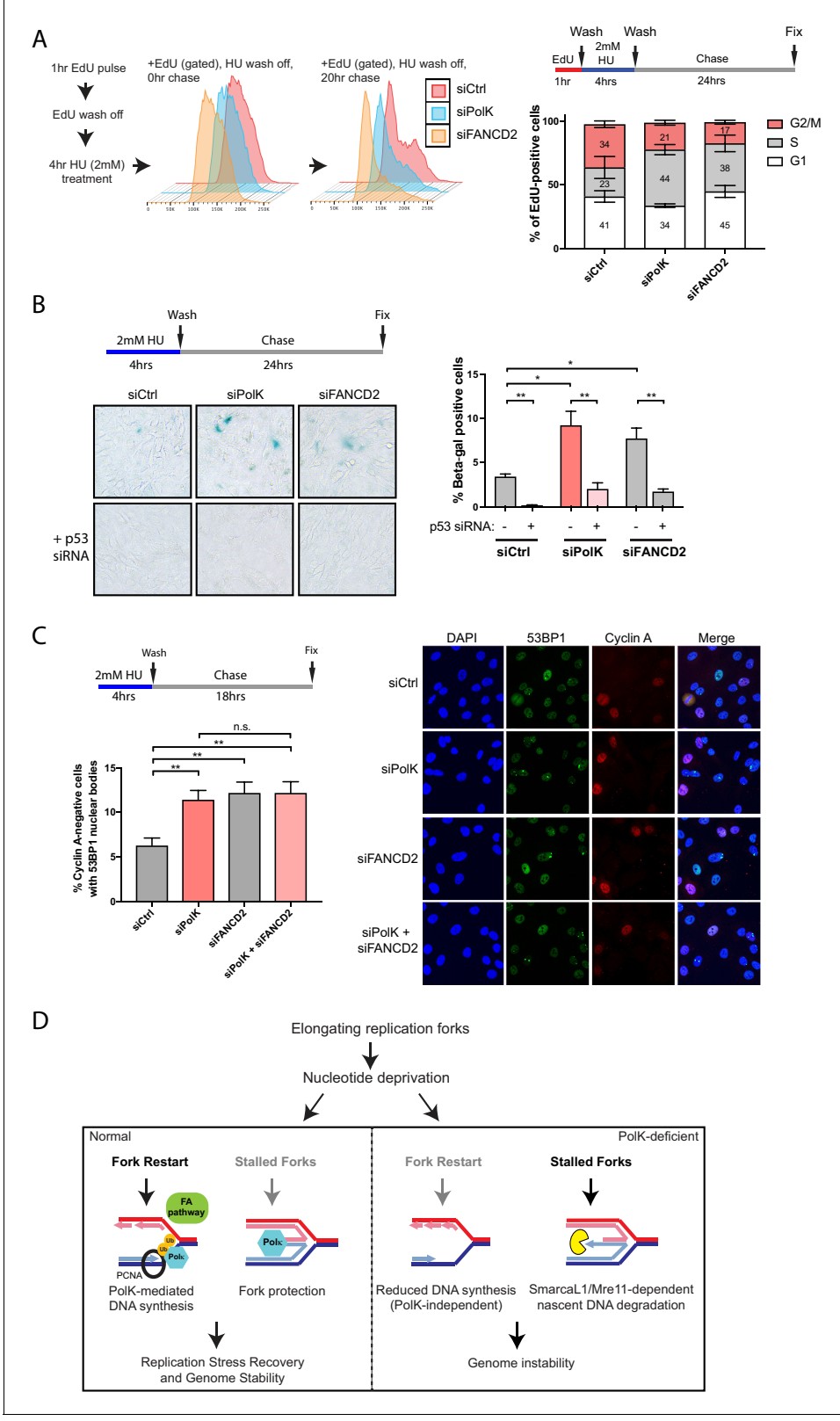

**Figure 6.** Loss of PolK leads to a p53-dependent cell cycle delay and 53BP1 nuclear body accumulation following HU pulse treatment. (**A**) Schematic for measuring cell cycle progression after recovery from HU pulse treatment (**A–C**). RPE-1 cells treated with the indicated siRNAs were initially pulsed-labeled with EdU (10 μM) for 1 hr to label untreated S-phase cells, followed by a wash step, an HU (2 mM) pulse treatment for 4 hr, another wash step, and recovery (chase) with fresh media for the indicated time. Recovery of EdU-positive, HU pulse-treated cells were tracked by FACS analysis and the
*Figure 6 continued on next page*

*Figure 6 continued*

proportion of cells in different cell cycle phases were determined by DAPI DNA content (FlowJo). Data represented from three independent experiments with mean ± s.d (**A**). (**B**) RPE-1 cells treated with the indicated siRNAs were treated with HU (4 hr), followed by a wash step, and chase for 24 hr with fresh media. Cells were then fixed and stained for SA-β-Gal activity. Data represented from three independent experiments with mean ± s.d., p-value calculated using t-test with Welch's correction. (**C**) RPE-1 cells treated with the indicated siRNAs were treated with HU (4 hr), followed by a wash step, and chase for 18 hr with fresh media. Cells were then fixed and co-stained for Cyclin A and 53BP1. Only Cyclin A-negative RPE-1 cells (G1 phase) were quantified for 53BP1 nuclear bodies. Data represented from three independent experiments with a minimum of 300 Cyclin A-negative cells per experiment; mean ± s.d. was plotted and p-value calculated using t-test with Welch's correction. (**D**) A model depicting how PolK promotes replication stress recovery and genome stability in an FA pathway-dependent manner in response to conditions of nucleotide starvation. n.s. = no significance, * = p < 0.05, ** = p < 0.01, *** = p < 0.001, **** = p < 0.0001.

DOI: https://doi.org/10.7554/eLife.41426.016

The following source data and figure supplement are available for figure 6:

**Source data 1.** Source data for *Figure 6C and B*, *Figure 6—figure supplement 1B and C*.
DOI: https://doi.org/10.7554/eLife.41426.018
**Figure supplement 1.** Mirin treatment does not rescue genome instability in HU pulse-treated PolK- or FANCD2-deficient cells.
DOI: https://doi.org/10.7554/eLife.41426.017

# Discussion

Studies have indicated the existence of tumorigenesis barriers that slow or inhibit the progression of preneoplastic lesions to neoplasia. One such barrier involves oncogene-induced DNA replication stress, which leads to activation of the DNA damage checkpoint, cell cycle arrest and/or senescence and apoptosis (*Schoppy et al., 2012*; *Gilad et al., 2010*; *Murga et al., 2011*). We propose that early deregulation of the replication stress response could lead to sub-lethal genomic instability that becomes progressively worse in subsequent rounds of cell division as cells escape oncogene-induced senescence. Thus, it is critical to understand the different mechanisms that govern replication fork recovery upon exposure to replication stress conditions. Based on our studies, we predict that a wide range of nucleotide starvation conditions, including oncogene-induced nucleotide exhaustion, could lead to a dependence on PolK and/or the FA pathway to help mammalian cells withstand limited nucleotide availability via mechanisms involving fork restart. Interestingly, a recent study by Vaziri and colleagues showed that PolK confers tolerance to oncogenic replication stress (*Yang et al., 2017*). Thus, we postulate that under certain conditions of low dNTP levels, the normal replicative polymerases (delta, epsilon) are disfavored in their usage while TLS Pols can readily engage at the stalled replication fork due to the low nucleotide concentration. This concept was originally proposed by Walker and colleagues, showing that the bacterial Y-family DNA polymerases Pol IV (DinB) and Pol V (UmuD$_2$'C) enhance cell survival under conditions of nucleotide starvation (*Godoy et al., 2006*). Additionally, PolK has been shown to be intrinsically more processive and has higher fidelity than the other Y-family TLS Pols (*Lone et al., 2007*). The stabilization of the replisome by PolK during HU treatment provides at least two major advantages: 1) fork stalling events that lead to reversed fork structures would remain protected against MRE11-dependent nuclease digestion of nascent DNA, and 2) a moving fork (albeit slower-moving) provides a reasonable temporary solution or surrogate prior to the eventual resumption of normal DNA synthesis for fork restart after nucleotide levels rise again, leading to a more efficient fork recovery process (*Figure 6D*). We speculate that the usage of PolK and other related processes may not be entirely stochastic, but are likely invoked at vulnerable regions of the genome. Uncovering the different mechanisms of fork recovery may help provide a clearer picture of how timely completion of DNA replication at the genome-wide level is required to prevent genome instability, and how these factors, including PolK and the FA pathway, can act as barriers against early stage carcinogenesis.

Previous studies have implicated both TLS Pol Eta (PolH) and PolK to complete DNA replication at common fragile sites (CFSs) (*Barnes et al., 2017*; *Rey et al., 2009*). A recent work from Nussenzweig and colleagues mapped genome-wide DNA break sites in response to prolonged HU-mediated replication stress in mammalian cells (*Tubbs et al., 2018*). Remarkably, they found that long poly (dA:dT) DNA tracts are preferred sites of replication fork stalling and collapse within early-replicating fragile sites (ERFSs) and late-replicating common fragile sites (CFSs). Intriguingly, they discovered a PolK mutational signature (*Hile and Eckert, 2008*) at these fragile poly(dA:dT) DNA structures, suggesting that throughout evolution, PolK has been employed during replication of these poly(dA:dT)

repeats at these replication fork-stalling sites. Thus, in our study, we provide a mechanistic insight into how PolK is recruited and utilized at stalled replication forks to aid in fork recovery. Implications of PolK's role in both genome maintenance and mutagenesis will be highly relevant in recurrent chromosomal rearrangements that arise from breakage within fragile hotspot regions throughout cancer genomes, as well as progression of preneoplastic lesions to genomicaly unstable cancers.

# Materials and methods

### Key resources table

| Reagent type (species) or resource | Designation | Source or reference | Identifiers | Additional information |
|---|---|---|---|---|
| Cell line (*Homo sapiens*) female | hTERT RPE-1 (RPE) | ATCC | CRL-4000 | |
| Cell line (*Homo sapiens*) female | U-2 OS (U2OS) | ATCC | HTB-96 | |
| Cell line (*Homo sapiens*) female | U2OS PolK KO (U2OS sgPolK Clone #1) | This Paper | | CRISPR-Cas9 generated using pX330 construct and guide RNAs listed in Materials and methods. |
| Cell line (*Homo sapiens*) | 293T | ATCC | CRL-3216 | |
| Cell line (*Homo sapiens*) | 293T PolK KO #1 (sgPolK Clone #1) | This Paper | | CRISPR-Cas9 generated using pX330 construct and guide RNAs listed in Materials and methods. |
| Cell line (*Homo sapiens*) | 293T PolK KO #2 (sgPolK Clone #2) | This Paper | | CRISPR-Cas9 generated using pX330 construct and guide RNAs listed in Materials and methods. |
| Cell line (*Homo sapiens*) | PD20 Vector | PMID: 11239454 | | *Garcia-Higuera et al., 2001* |
| Cell line (*Homo sapiens*) | PD20 FANCD2 WT | PMID: 11239454 | | *Garcia-Higuera et al., 2001* |
| Cell line (*Homo sapiens*) | PD20 FANCD2 K561R | PMID: 11239454 | | *Garcia-Higuera et al., 2001* |
| Transfected construct (*Homo sapiens*) | pCDNA3.1 HA-PCNA WT | PMID: 22157819 | | *Dungrawala and Cortez, 2015* |
| Transfected construct (*Homo sapiens*) | pCDNA3.1 HA-PCNA K164R | PMID: 22157819 | | *Dungrawala and Cortez, 2015* |
| Recombinant DNA reagent | eGFP-C1 | ClonTech | | |
| Transfected construct (*Homo sapiens*) | eGFP-C1-PolK WT | PMID: 22157819 | | *Dungrawala and Cortez, 2015* |
| Transfected construct (*Homo sapiens*) | eGFP-C1-PolK CD | This Paper | | Site directed mutagenesis using primers listed in Materials and methods. D198A/E199A |

*Continued on next page*

Continued

| Reagent type (species) or resource | Designation | Source or reference | Identifiers | Additional information |
|---|---|---|---|---|
| Transfected construct (*Homo sapiens*) | eGFP-C1-PolK UBZ | This Paper | | Site directed mutagenesis using primers listed in Materials and methods. D644A/D799A |
| Transfected construct (*Homo sapiens*) | eGFP-C1-PolH WT | *Dungrawala and Cortez, 2015* *Dungrawala and Cortez, 2015* PMID: 22157819 | | |
| Transfected construct (*Homo sapiens*) | eGFP-C1-PolI WT | *Dungrawala et al., 2015* PMID: 22157819 | | |
| Recombinant DNA reagent | pX330 | Addgene | #42230 | |
| Antibody | Mouse anti-IdU | BD Bioscience | Cat. #: 347580 | IF (1:200) |
| Antibody | Rat anti-CldU | Abcam | Cat. #: ab6326 | IF (1:100) |
| Antibody | Mouse anti-PCNA | Abcam | Cat. #: ab29 | WB (1:5000) |
| Antibody | Rabbit anti-PCNA-Ub | Cell Signaling | Cat. #: D5C7P | WB (1:1000) |
| Antibody | Mouse anti-FANCD2 | Santa Cruz | Cat. #: sc-20022 | WB (1:500) |
| Antibody | Rabbit anti-FANCD2 | Novus Biological | Cat. #: NB100-182 | WB (1:5000) |
| Antibody | Rabbit anti-pCHK1 S345 | Cell Signaling | Cat. #: 133D3 | WB (1:5000) |
| Antibody | Goat anti-CHK1 | Abcam | Cat. #: ab2845 | WB (1:5000) |
| Antibody | Mouse anti-gH2AX | EMD Millipore | Cat. #: 05–636 | WB (1:5000) |
| Antibody | Mouse anti-GFP | Santa Cruz | Cat. #: sc-9996 | IF (1:200), WB (1:5000) |
| Antibody | Mouse anti-Cyclin A2 | CalBiochem | clone E23 | IF (1:100) |
| Antibody | Rabbit anti-53BP1 | Abcam | Cat. #: ab21083 | IF (1:200) |
| Antibody | Mouse anti-POLK | Santa Cruz | Cat. #: sc-166667 | WB (1:1000) |
| Antibody | Mouse anti-POLH | Santa Cruz | Cat. #: sc-17770 | WB (1:1000) |
| Antibody | Rabbit anti-POLI | Bethyl | Cat. #: A301-304A | WB (1:5000) |
| Antibody | Rabbit anti-MCM2 | Bethyl | Cat. #: A300-094A | WB (1:5000) |
| Antibody | Rabbit anti-MCM5 | Bethyl | Cat. #: A300-195A | WB (1:5000) |
| Antibody | Rabbit anti-Histone H3 | Abcam | Cat. #: ab1791 | WB (1:5000) |
| Antibody | Rabbit anti-REV7 | Abcam | Cat. #: ab180579 | WB (1:5000) |
| Antibody | Rabbit anti-pRPA32 S33 | Bethyl | Cat. #: A300-246A | WB (1:5000) |

*Continued on next page*

*Continued*

| Reagent type (species) or resource | Designation | Source or reference | Identifiers | Additional information |
|---|---|---|---|---|
| Antibody | Rabbit anti-pRPA32 S4/S8 | Bethyl | Cat. #: A700-009 | WB (1:5000) |
| Antibody | Rabbit anti-RPA32 | Bethyl | Cat. #: A300-244A | WB (1:5000) |
| Antibody | Mouse anti-BRCA2 | CalBiochem | clone 2B | WB (1:1000) |
| Antibody | Mouse anti-HA | BioLegend | Cat. #: 901502 | WB (1:5000) |
| Antibody | Goat anti-HisTag | Bethyl | Cat. #: A190-113A | WB (1:5000) |
| Antibody | Rabbit anti-REV1 | Santa Cruz | Cat. #: sc-48806 | WB (1:1000) |
| Antibody | Rabbit anti-FANCI | Bethyl | Cat. #: A301-254A | WB (1:5000) |
| Antibody | Rabbit anti-MCM4 | Bethyl | Cat. #: A300-193A | WB (1:5000) |
| Antibody | Rabbit anti-MCM3 | Bethyl | Cat. #: A300-192A | WB (1:5000) |
| Antibody | Rabbit anti-Rad51 | Abcam | Cat. #: ab63801 | WB (1:5000) |
| Antibody | Rabbit anti-pRPB1 CTD S2 | Cell Signaling | Cat. #: E1Z3G | WB (1:1000) |
| Antibody | Mouse anti-RPB1 CTD | Cell Signaling | Cat. #: 2629 | WB (1:1000) |
| Antibody | Rabbit anti-pCHK2 T68 | Cell Signaling | Cat. #: C13C1 | WB (1:5000) |
| Recombinant DNA reagent | Fugene 6 Transfection reagent | Promega | E2692 | |
| Recombinant DNA reagent | Lipofectamine RNAiMAX | Invitrogen | Cat #: 13778150 | |
| Sequence-based reagent | CRISPR guide RNAs (sgPolK#1,#2) | This Paper | | See Materials and methods |
| Sequence-based reagent | siRNAs | This Paper | | See Materials and methods |
| Sequence-based reagent | Mutagenesis primers | This Paper | | See Materials and methods |
| Peptide, recombinant protein | Tri-Ubiquitin chains (K48-linked) | Boston Biochem | UC-215B | |
| Peptide, recombinant protein | Tri-Ubiquitin chains (K63-linked) | Boston Biochem | UC-315B | |
| Peptide, recombinant protein | USP2 Catalytic Domain (CD) | Boston Biochem | E-504 | |
| Peptide, recombinant protein | AMSH | Boston Biochem | E-548B | |
| Peptide, recombinant protein | SARS PLPro | Boston Biochem | E-610 | |
| Commercial assay or kit | Click-it EdU Imaging Kit | Invitrogen | C10339 | |

*Continued on next page*

*Continued*

| Reagent type (species) or resource | Designation | Source or reference | Identifiers | Additional information |
|---|---|---|---|---|
| Commercial assay or kit | Click-it EdU Flow Cytometry Assay Kit | Invitrogen | C10646 | |
| Commercial assay or kit | QuikChange XL Mutagenesis Kit | Agilent | Cat #: 200517 | |
| Commercial assay or kit | TOPO-TA Cloning Kit for Sequencing | Invitrogen | Cat #: 450071 | |
| Chemical compound, drug | Hydroxyurea (HU) | Sigma | H8627 | |
| Chemical compound, drug | Aphidicolin (APH) | Sigma | A4487 | |
| Chemical compound, drug | Gemcitabine (Gem) | Sigma | G6423 | |
| Chemical compound, drug | Mirin | Sigma | M9948 | |
| Chemical compound, drug | AZD7762 (CHK1 inhibitor) | Sigma | SML0350 | |
| Chemical compound, drug | 1-b-D-ribofuranoside (DRB) | Sigma | D1916 | |
| Chemical compound, drug | Flavopiridol (FVP) | Sigma | F3055 | |
| Chemical compound, drug | Formaldehyde 37% w/v | VWR | M134 | |
| Chemical compound, drug | Glycine | Fischer Scientific | BP381 | |
| Chemical compound, drug | 5'-Iodo-2'-deoxyuridine (IdU) | Sigma | I7125 | |
| Chemical compound, drug | 5'-chloro-2'-deoxyuridine (CldU) | Sigma | C6891 | |
| Chemical compound, drug | 5'-ethynyl-2-deoxyuridine (EdU) | Sigma | Cat #: 900584 | |
| Chemical compound, drug | Thymidine (dT) | Sigma | T1895 | |
| Chemical compound, drug | 2'-deoxycytidine HCl (dC) | Sigma | D0776 | |
| Chemical compound, drug | 2'-deoxyadenosine (dA) | Sigma | D8668 | |
| Chemical compound, drug | 2'-deoxyguanosine (dG) | Sigma | D0901 | |
| Chemical compound, drug | Cytidine (rC) | Sigma | C4654 | |
| Chemical compound, drug | Adenosine (rA) | Sigma | A4036 | |
| Chemical compound, drug | Guanosine (rG) | Sigma | G6264 | |
| Chemical compound, drug | Uridine (rU) | Sigma | U3003 | |
| Chemical compound, drug | Anti-GFP mAb agarose beads | MBL | Cat #: D153-8 | |
| Chemical compound, drug | Dynabeads Myone Streptavidin T1 | ThermoFisher Scientific | Cat #: 65601 | |

*Continued on next page*

*Continued*

| Reagent type (species) or resource | Designation | Source or reference | Identifiers | Additional information |
|---|---|---|---|---|
| Chemical compound, drug | Biotin Azide | ThermoFisher Scientific | Cat #: B10184 | |
| Chemical compound, drug | Protein G beads agarose | ThermoFisher Scientific | Cat #: 20399 | |
| Chemical compound, drug | SYPRO Ruby Protein Gel Stain | ThermoFisher Scientific | Cat #: S12000 | |
| Chemical compound, drug | cOmplete Mini Protease Inhibitor Cocktail | Sigma | Cat #: 11836170001 (Roche) | |
| Software, algorithm | GraphPad Prism | (https://graphpad.com) | RRID:SCR_015807 | |
| Software, algorithm | ImageJ | (https://imagej.nih.gov/ij/) | RRID:SCR_003070 | |
| Software, algorithm | FlowJo | (https://www.flowjo.com/) | RRID:SCR_008520 | |

## Cell culture

U2OS and 293 T cells (ATCC) were grown at 37°C in DMEM (Gibco) with 10% FBS (Atlantic Biologicals), 1% penicillin/streptomycin (Gibco), and 1% glutamine (Gibco). hTERT-immortalized RPE-1 cells (ATCC) were grown in DMEM/F12 (Gibco) with 10% FBS, 1% penicillin/streptomycin,. 25% Sodium Bicarbonate (Gibco). PD20 patient cells (*Garcia-Higuera et al., 2001*; *Timmers et al., 2001*) were grown in DMEM with 15% FBS, 1% penicillin/streptomycin and 1% glutamine. All cell lines were tested for mycoplasma using the Roche MycoTOOL detection kit. All cell lines, except for the PD20 cells, were also authenticated by short tandem repeat (STR) profiling.

## DNA fiber analysis

DNA fibers were prepared as described previously (*Chen et al., 2015*). Briefly, cells were pulsed with 50 µM IdU and CldU for times indicated in each experiment. After trypsinization, cells were washed and resuspended at $1 \times 106$/mL in cold PBS, 2 uL were plated onto a glass slide, and lysed with 10 uL lysis buffer (0.5% SDS, 200 mM Tris-HCl pH 7.4, 50 mM EDTA) for 6 min. Slides were tilted at a 15 degree angle to allow DNA spreading, and then fixed for 3 min in chilled 3:1 methanol: acetic acid. The DNA was denatured with 2.5 N HCl for 30 min, washed in PBS, blocked for 1 hr in 5% BSA in PBS with 0.1% Triton X-100. Slides were stained for 1 hr with primary antibodies, washed 3X in PBS, stained for 30 min with secondary antibodies, washed 3X in PBS and dried. Coverslips were mounted with Prolong antifade reagent and sealed. Slides were imaged with Keyence BZ-X710 microscope. Image analysis was done with ImageJ. A minimum of 150 fibers were measured for each independent experiment for percentage of restart, and analysis shows mean of three independent experiments. A minimum of 60 fiber lengths were measured for each independent experiment measuring tract length, and analysis shows the pool of three independent experiments (biological replicates). Tract lengths were calculated by converting µm measured in ImageJ to kb using the conversion 1 µm = 2.59 kb (*Ray Chaudhuri et al., 2016*).

## Western blotting

Western blots were performed with whole-cell extracts prepared in SDS sample buffer (0.1M Tris pH 6.8, 2% (w/v) SDS and 12% (v/v) β-mercaptoethanol). Protein extracts were separated on Nupage 4–12% Bis-Tris or 3–8% Tris-Acetate gels (Invitrogen). Proteins were transferred onto 0.45 µM PVDF membrane in Invitrogen Tris-Glycine transfer buffer. Membranes were blocked in 5% milk in TBST for 1 hr and incubated in primary antibody overnight. Next day membranes were incubated with secondary antibodies in 5% milk TBST for 2 hr at room temperature and developed using Western Lightning Plus-ECL reagent. For detection of BRCA2 protein, cells were lysed directly on the plate with ice-cold NP-40 lysis buffer (50 mM HEPES pH 7.5, 250 mM NaCl, 1% NP-40, 0.5 mM EDTA, 1 mM DTT, supplemented with Roche protease inhibitor cocktail) for 10 min on ice, scraped and

transferred into an Eppendorf tube, and spun at 4°C for 10 min at 14 k rpm. Lysates were heated to 54°C for 4 min prior to loading onto a 3–8% Tris-Acetate gel, and all subsequent steps were performed as described above.

## Cross-linking Immunoprecipitation

Crosslinking protocol was performed as described in *Kannouche and Lehmann (2006)*, but with slight modifications. Briefly, one 10 cm plate of cells per sample was washed once in PBS, cross-linked for 10 min with 3 mL 1% formaldehyde in PBS, and terminated with 300 µL of 1.25M glycine. Cells were then scraped and transferred into a falcon tube, washed 3X in cold PBS, and resuspended in 250 µL lysis buffer (50 mM Tris pH 7.5, 150 mM NaCl, 0.3% SDS) and lysed for 10 min at RT. Samples were sonicated using microtip sonicator as follows: 3 pulses at 33% amplitude for 12 s with 30 s on ice in between, and one final pulse at 40% amp for 12 s. Lysates were centrifuged at high speed for 5 min, transferred to a new tube, and diluted 1:8 in dilution buffer (50 mM Tris pH 7.5, 150 mM NaCl, 5 mM EDTA, 0.1% Triton). Samples were then incubated at 4°C overnight with antibody. The following day, samples were incubated for 1 hr at 4°C with 20 µL of 50% slurry Protein G beads. Beads were then washed 4X in 500 µL dilution buffer, and boiled for 15 min in 4X Lamelli buffer supplemented 100 mM DTT. GFP-IPs were performed using GFP-agarose beads incubated overnight in place of antibody, with the subsequent 4X washes performed the next day.

## iPOND

iPOND experiments were performed as described in *Dungrawala et al. (2015)*. In brief, three 15 cm plates of 293 T cells were prepared per sample, and treated for 10 min with 10 µM EdU, or pre-treated for 1 hr with 2 mM HU incubation followed by 4 hr with 2 mM HU and 10 µM EdU. Chases were performed for 1 hr with 10 µM thymidine. Plates were cross-linked with 10 mL 1% formaldehyde in PBS for 20 min and terminated with 1 mL 1.25M glycine. Cells were scraped and transferred into a 50 mL falcon tube, washed 3X in PBS. Cells were resuspended in 0.25% Triton for 30 min at RT, and washed once in 0.5% BSA in PBS and once in PBS. Cells were resuspended in click reaction (2 mM CuSO4, 10 mM sodium ascorbate, and 10 µM biotin azide in PBS) for 2 hr at RT and then washed once in 0.5% BSA in PBS and once in PBS. Cell pellets were resuspended in 1 mL lysis buffer (50 mM Tris pH 8, 1% SDS with protease inhibitor (Roche)), and sonicated for 20 s at 40% amplitude with 1 min on ice for a total of 5 pulses. After centrifugation, lysates were transferred to a new tube and diluted 1:1 in PBS. Next, 25 µL of streptavidin magnetic beads (Thermo Fisher) (washed 3X in PBS) were added and incubated overnight at 4°C. Beads were washed in 1 mL of cold lysis buffer, 1 mL of 1M NaCl, and then two more washes with cold lysis buffer. Proteins were eluted with addition of 2X SDS loading buffer (5% SDS, 25% glycerol, 0.15M Tris pH 6.8, 200 mM DTT) and boiled at 95°C for 25 min. Samples were then loaded into NuPage gels and western was performed as described.

## Ubiquitin chain restriction and ubiquitin cleavage assays

Protocol for ubiquitin chain cleavage assays was performed as described previously (*Békés et al., 2015*). In vitro reactions were performed in 150 mM NaCl, 20 mM Tris pH 8 using 0.5 µg ubiquitin chains (Boston Biochem). Reactions with SARS PLpro or USP2 Catalytic Domain (CD) also contained 5 mM DTT. Samples were incubated with 100 ng of each DUB at 37°C for the times indicated. Reactions were terminated by addition of 4X Lamelli buffer and boiling for 10 min. Samples were loaded onto NuPage 4–12% Bis Tris gels, stained using SYPRO Ruby gel stain (BioRad), and imaged on Bio-Rad EZ Gel Doc detector. For cross-linked IP samples, beads were washed 3X in dilution buffer, and then buffer exchanged with 4 × 500 µL washes in reaction buffer above. Reactions were then performed for 1 hr at 37°C with 900 ng of enzyme and terminated similarly.

## Immunofluorescence

EdU imaging was performed according to Click-It EdU Imaging Kit protocol (Invitrogen) with slight modifications. Cells were grown on coverslips and fixed in 3.7% formaldehyde for 15 min at RT, and washed with 3% BSA in PBS. Coverslips were permeabilized in 0.5% Triton X-100 for 20 min at RT and washed twice in 3% BSA. Coverslips were incubated in click reaction buffer for 30 min at RT and washed once in 3% BSA. For subsequent antibody incubations, coverslips were blocked for 1 hr in

2%BSA, 0.2% Triton at RT, and incubated for 2 hr at RT with primary antibody. Coverslips were washed 2X in blocking buffer, incubated for 45 min secondary antibody, washed 2X in blocking buffer and then once in PBS. Coverslips were mounted onto glass slides using Vectashield with DAPI. For regular immunofluorescence, cover slips were fixed for 10 min in 3.7% formaldehyde (most antibodies), or 10 min in 3.7% formaldehyde followed by a 5 min ice-cold methanol fixation (Cyclin A staining). Cover slips were permeabilized for 10 min with 0.5% Triton-X 100, and blocking and staining was performed similarly as described.

## Single-molecule localization microscope

The Single-molecule localization imaging was performed on a customized inverted microscope as described previously (*Chen et al., 2015*). In brief, the 640 nm (UltraLaser, MRL-FN-639–800), 561 nm (UltraLaser, MGL-FN-561–200), and 488 nm (OBIS) laser lines were collimated and reflected by a penta-edged dichroic beam splitter (FF408/504/581/667/762-Di01) into an TIRF Objective (HCX PL APO 63X NA = 1.47, Zeiss). The illuminations were adjusted into a HILO mode and tuned to ~0.8, 1.0, and 1.5 kW/cm$^2$, respectively, for nucleus imaging. The emitted fluorescence was further magnified by a 2X lens tube (Diagnostic Instruments) before collected onto am sCMOS camera (Prime 95B, Photometrics). After the 2X magnification, the emitted fluorescence was filtered by a single-band filter (Semrock FF01-676/37 for Alexa Fluor 647 for EdU detection in *Figure 5*, and FF01-809/81 and FF01-607/36 for Alexa Fluor 488 and 568, respectively, for GFP-PolK and PCNA detection in *Figure 5—figure supplement 3A*, in which the multiplexed imaging was accomplished by sequentially switching the illumination laser and such filters accordingly). The photons were then recorded at 33 Hz for 2000 frames for each imaging. Each image taken from different color channels (*Figure 5—figure supplement 3A*) were mapped using a 2nd polynomial mapping algorithm. In brief, before each imaging, the broad-spectrum fluorescent beads (TetraSpec, Thermo Fisher) were imaged in different color-channels, and the mass centers recorded in each channel were submitted for 2nd polynomial regression, which then optimized the 2nd polynomial's ecoefficiency for mapping the nucleus images.

## Single-molecule localization

After collecting 200 frames for each imaging, the single-molecule image stack was submitted to a home-built software for precise single-molecule localization. In brief, each frame from an image stack was box filtered, roughly-local-maxima-localized, segmented, and submitted to GPU for parallel fitting each single Point-Spread-Function (PSF) using the Maximum Likelihood Estimation (MLE), and the fitting accuracy was evaluated using Cramer-Rao Lower Bound (CRLB) estimation. We note that the patterned read-out noise of the sCMOS camera was calibrated before imaging. Such read-out noise for each pixel was approximated into a Gaussian distribution and contributed to the MLE fitting for each single PSF (*Rust et al., 2006*). The coordinates were then submitted for Auto-Pair-Correlation (*Figure 5d and e*) *Figure 5D, E* and Cross-Pair-Correlation (*Figure 5—figure supplement 3A*). The Pair-Correlation function uses the molecular coordinates obtained from the single-molecule localization experiments to define the average probability density of finding fluorophores with specific molecular proximities around each fluorophore. By integrating the probability density profile of each cell and multiplying with the average density of fluorophores detected in the nucleus, we can derive the average number of fluorophores within a cluster (*Sengupta et al., 2011*). Since EdU was ~1:1 tagged by fluorophores via click chemistry, such number of fluorophores per cluster is proportional to the number of EdU molecules per focus. Using these algorithms, we can arrive at the number of EdU molecules per focus and estimate the number of EdU foci per nucleus by dividing the relative total number of EdU in each nucleus by the average number of EdU per focus. The statistics of *Figure 5D* is mean ± s.d., N = 59, 43, 38, and 55 nuclei for siCtrl, siPolK, siFANCD2, and siFANCD2 +siPolK, respectively. For *Figure 5E* and *Figure 5—figure supplement 3A*, the mean ±s. d., N = 15, 21, 18, and 18 nuclei for WT_NT, CD_NT, WT_HU, and CD_HU, respectively.

## Flow cytometry

Protocol was performed according to Click-It EdU Flow Cytometry Imaging Kit (Invitrogen). Cells were labeled with 10 µM EdU for 1 hr prior to HU treatment and subsequent chases. Cells were fixed for 15 min in 3.7% formaldehyde, washed in 3% BSA, and resuspended in Saponin buffer. Cells

were incubated with click reaction buffer for 30 min at RT and washed once in saponin based wash reagent. Pellets were resuspended in 200 µL and incubated with primary antibody for 2 hr. Pellets were washed twice in 3 mL saponin wash buffer, and incubated for 1 hr with secondary antibody and washed once. Pellets were then resuspended in 500 µL of saponin buffer containing 1 mg/mL DAPI with RNase A and incubated for 1 hr at 37°C. Cells were analyzed with LSRII UV flow cytometer and analysis done using FlowJo software.

## Senescence-associated β galactosidase staining

Protocol was performed as described (*Lau and David, 2017*). Briefly, cells were plated in 12-well or 6-well plates. Following 4 hr HU treatment and chase, cells were fixed with 2% formaldehyde, 0.2% glutaraldehyde for 10 min, washed in PBS, and incubated overnight in humidified 37°C chamber in SA-βGal staining solution (25.2 mM sodium phosphate dibasic, 7.34 mM citric acid at pH 6, 150 mM NaCl, 2 mM MgCl2, 5 mM potassium ferricyanide, 5 mM potassium ferrocyanide, and 1 mg/mL X-gal). Following morning cells were washed 3X in water, stained with DAPI, and imaged with bright-field microscopy.

## Plasmids, primers, siRNAs, and CRISPR sgRNAs

Transient plasmid transfections were performed using FuGene six reagent (Promega), and siRNA transfections were performed using Lipofectamine RNAiMax (Invitrogen), both according to manu-facturer's instructions. Analyses were done between 24–48 hr after plasmid transfection and 72 hr after siRNA transfection. CRISPR generation was performed using the pX330 system from the Feng Zhang lab with puromycin resistance cloned by the Agnel Sfeir lab (gift). Sanger sequencing of CRISPR clones was performed on gDNA fragments cloned into TOPO TA vector (Thermo Fisher). Site directed mutagenesis was performed using Agilent xL kit according to manufacturer's instructions. TLS constructs are in eGFP-C1 vectors (Clontech). PolK CD (catalytic dead mutant)=D198A/E199A, PolK UBZ (double UBZ mutant)=D644A/D799A. The siRNA-resistant HA-PCNA WT and K164R constructs were cloned in pcDNA3.1 vector, and was generated previously (*Jones et al., 2012*). siRNA target sequences (Qiagen) used in this study:

*PolK* siRNA (siPolK): #1, 5'-TGGAATTAGAACAAAGCCGAA-3',
#2, 5'-AACCTCTAGAAATGTCTCATA-3'
*PolH* siRNA (siPolH): #1, 5'-ATCCATTTAGGTGCTGAGTTA-3', #2, 5'-CAGCCAAATGCCCA TTCGCAA-3'
*PolI* siRNA (siPolI): #1, 5'-GCGGTTTATTAAGCTCTTCTA-3', #2, 5'-TTCGGATTAGCGGTTTA TTAA-3'
*Rev1* siRNA (siRev1): #1, 5'-CAGCGCATCTGTGCCAAAGAA-3', #2, 5'-CTGCCAGGTCCAAGCAATATA-3'
*MAD2L2* siRNA (siRev7): #1, 5'-GTGGAAGAGCGCGCTCATAAA-3', #2, 5'-AAGATGCAGC TTTACGTGGAA-3'
*REV3L* siRNA (siRev3L): #1, 5'- CGGGATGTAGTCAAACTGCAA-3', #2, 5'- ATGAGTATGGATCA TATACAA-3'
*PCNA* siRNA (siPCNAresist): 5'- GCCGAGATCTCAGCCATATTT-3'
*TP53* siRNA (siP53): 5' CAGAGTGCATTGTGAGGGTTA-3'
*FANCD2* siRNA (siFANCD2): 5'- AACAGCCATGGATACACTTGA-3'
*RAD51* siRNA (siRad51): #1, 5'- CAGGATAAAGCTTCCGGGAAA-3', #2, 5'- CACTTCTAAATTAA TGGTAAA-3'
*BRCA2* siRNA (siBRCA2): #1, 5'- CAGCGTTTGTGTATCGGGCAA-3', #2, 5'- TACGTACTCCA-GAACATTTAA-3'
*SMARCAL1* siRNA (siSMARCAL1): 5'-TTGAGTTATGAGTTAGGTCAA–3' sgPolK guides: #1, 5' GAAGACTCATGGCCATAAAAT-3', #2, 5'- GATTGGCCTGAGGATAAAAGA-3'
Primers for site-directed mutagenesis:
PolK CD: fwd: 5'-CAATTTTATGGCCATGAGTCTTGCTGCAGCCTACTTGAATATAACAAAGC-3'
PolK UBZ1: fwd, 5'- CCTTGAATAAACATGTAGCCGAATGTCTTGATGGACC-3'
PolK UBZ2: fwd, 5'-CTGTTCAATGTGCATGTGGCCGTTTGCTTAAATAAAAG-3'

## Acknowledgements

We thank M Bekes (Nurix), M Jones (MSKCC), A Sfeir (NYU), K Coleman (NYU) and Y-H Chen (Miroculus) for technical assistance and guidance on CRISPR-Cas9-mediated generation of PolK sgRNA clonal cells, DNA fiber analysis, purified recombinant DUBs, and DUB cleavage assays, and members of the Huang laboratory for critical discussion. PT was supported, in part, by NIH grant T32 GM115313. This work was supported by NIH grants ES025166 (TTH) and GM108119 (ER), and ACS grant RSG-16-241-01-DMC (ER). Funds from the Basser Center for BRCA research from UPenn also supported TTH and the V Foundation for BRCA Cancer Research to TTH and ER.

## Additional information

### Funding

| Funder | Grant reference number | Author |
|---|---|---|
| National Institutes of Health | T32GM115313 | Peter Tonzi |
| American Cancer Society | RSG-16-241-01-DMC | Eli Rothenberg |
| National Institutes of Health | GM108119 | Eli Rothenberg |
| V Foundation for Cancer Research | | Eli Rothenberg<br>Tony T Huang |
| Basser Centre for BRCA | | Tony T Huang |
| National Institutes of Health | ES025166 | Tony T Huang |

The funders had no role in study design, data collection and interpretation, or the decision to submit the work for publication.

### Author contributions

Peter Tonzi, Conceptualization, Data curation, Formal analysis, Investigation, Methodology, Writing—review and editing; Yandong Yin, Data curation, Formal analysis, Methodology; Chelsea Wei Ting Lee, Data curation, Formal analysis; Eli Rothenberg, Supervision, Funding acquisition, Project administration; Tony T Huang, Conceptualization, Funding acquisition, Writing—original draft, Project administration, Writing—review and editing

### Author ORCIDs

Yandong Yin http://orcid.org/0000-0003-2499-871X
Tony T Huang http://orcid.org/0000-0001-9291-5002

### Decision letter and Author response

Decision letter https://doi.org/10.7554/eLife.41426.022
Author response https://doi.org/10.7554/eLife.41426.023

## Additional files

### Supplementary files

• Transparent reporting form
DOI: https://doi.org/10.7554/eLife.41426.019

### Data availability

All data generated or analysed during this study are included in the manuscript and supporting files.

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
