## [Decision Letter]

Thank you for submitting your article "Translesion polymerase kappa-dependent DNA synthesis underlies replication fork recovery" for consideration by *eLife*. Your article has been reviewed by three peer reviewers, and the evaluation has been overseen by a Reviewing Editor and Jessica Tyler as the Senior Editor. The following individual involved in review of your submission has agreed to reveal her identity: Kristin Eckert (Reviewer #3).

The reviewers have discussed the reviews with one another and the Reviewing Editor has drafted this decision to help you prepare a revised submission.

Summary:

In this manuscript, the authors demonstrate the role of translesion (TLS) polymerases in protecting and restarting stalled replication fork specifically under nucleotide deprived condition (i.e. hydroxyurea-induced stalling). Based on their initial screen of various TLS polymerases by employing the single-molecule DNA fiber technique coupled with siRNA strategy, the authors singled out PolK as the key polymerase responsible in promoting fork restart. This approach also revealed that PolK is epistatic to the Fanconi Anemia (FA) pathway in facilitating fork restart. In addition, through complementation approach, the authors demonstrated that both catalytic and PCNA-interacting domain of PolK is responsible in mediating fork restart; interestingly, only the PCNA-interacting domain is critically involved in preventing nascent DNA degradation. Collectively, this is a very interesting and well-conducted study that points to a previously unappreciated role of PolK in genome stability and replication stress recovery. However, several concerns need to be addressed, as indicated below.

Essential revisions:

1) One of the most salient issues brought up by two reviewers pertains to the major conclusion that nucleotide deprivation (through addition of HU) is the underlying cause of the reliance on PolK in fork restart and protection. Alternative pathways may contribute to replication stress by HU (for instance, transcriptional R loops cause replication stress, and DNA breaks in HU treated cells preferentially occur at genes transcriptionally induced by HU (Hoffman et al., 2015)). Moreover, HU is not a specific inhibitor of ribonucleotide reductase, and has been shown to induce oxidative stress in cells by affecting other Fe-S cluster enzymes (Huang et al., 2016). The reviewers are requesting the authors to provide additional experimental evidence that the phenotypes are due to nucleotide deprivation; by either:

a) Adding exogenous nucleosides to the culture media and measure whether the key molecular phenotypes are reversed (see Técher et al., 2016; Aird et al., 2013).

b) Utilizing a different stalling condition on some of the key data, for example aphidicholin which inhibits the DNA replication machinery without affecting the nucleotide pool level.

2) When discussing the results of Figure 4A, the authors state that "PolK depletion further reduced DNA synthesis (or fork speed) during HU treatment, as measured by reduced CldU tract length adjacent to IdU tracts". Could this reduction be due to the fact that PolK depleted cells have a defect in fork restart? During the time of HU treatment, replication forks will continuously stall and restart, and the reduced efficiency of restart associated with the loss of PolK could lead to shorter CldU tracts.

3) Figure 5. The authors should test how the cell cycle detects and the 53BP1 nuclear body formation accumulation observed upon PolK loss are affected by co-depletion of FANCD2 and PolK to confirm that these proteins act within the same pathway.

4) The authors show (Figure 2) that Pol kappa-deficiency does not affect Chk1 signaling in response to high dose HU treatment. This result directly contradicts a previous study (Bétous et al., 2013) and impacts ongoing discussions regarding reproducibility in science. This discrepancy is important for the field to reconcile. The authors should provide more information regarding possible differences in cell lines (genetic differences), doses, antibodies used, or other possible experimental reasons underlying differences.

5) Why does the expression of PolK UBZ mutant in the complementation experiment decrease the CldU track length as compared to vector (Figure 3B)? Is this statistically significant, and if so, what this may indicate mechanistically? It would be wise to add this in the Discussion section.

6) Figure 4D. Why FANCD2 depletion does not affect EdU foci counts while it increases the CldU tract lengths in Figure 4A?

References:

Aird KM, Zhang G, Li H, Tu Z, Bitler BG, Garipov A, Wu H, Wei Z, Wagner SN, Herlyn M, Zhang R. Suppression of nucleotide metabolism underlies the establishment and maintenance of oncogene-induced senescence.

Cell Rep. 3(4):1252-65. doi: 10.1016/j.celrep.2013.03.004. (2013)

Hoffman EA, McCulley A, Haarer B, Arnak R, Feng W. Break-seq reveals hydroxyurea-induced chromosome fragility as a result of unscheduled conflict between DNA replication and transcription. Genome Res. 25(3):402-12. doi: 10.1101/gr.180497.114. (2015)

Técher H, Koundrioukoff S, Carignon S, Wilhelm T, Millot GA, Lopez BS, Brison O, Debatisse M. Signaling from Mus81-Eme2-Dependent DNA Damage Elicited by Chk1 Deficiency Modulates Replication Fork Speed and Origin Usage.

Cell Rep. 14(5):1114-1127. doi: 10.1016/j.celrep.2015.12.093. (2016)

---

## [Author Response]

Essential revisions:1) One of the most salient issues brought up by two reviewers pertains to the major conclusion that nucleotide deprivation (through addition of HU) is the underlying cause of the reliance on PolK in fork restart and protection. Alternative pathways may contribute to replication stress by HU (for instance, transcriptional R loops cause replication stress, and DNA breaks in HU treated cells preferentially occur at genes transcriptionally induced by HU (Hoffman et al., 2015)). Moreover, HU is not a specific inhibitor of ribonucleotide reductase, and has been shown to induce oxidative stress in cells by affecting other Fe-S cluster enzymes (Huang et al., 2016). The reviewers are requesting the authors to provide additional experimental evidence that the phenotypes are due to nucleotide deprivation; by either:a) Adding exogenous nucleosides to the culture media and measure whether the key molecular phenotypes are reversed (see Técher et al., 2016; Aird et al., 2013).b) Utilizing a different stalling condition on some of the key data, for example aphidicholin which inhibits the DNA replication machinery without affecting the nucleotide pool level.

To address the concern of HU potentially having indirect (off target) effects, such as increased ROS that may affect the interpretation of our results, we have now modified the restart assay by supplementing deoxynucleosides in addition to CldU to the high dose HU (2mM) medium (under continuous HU treatment), in comparison to our standard restart assay where the HU is washed away and replaced with fresh media and CldU. Doing so resulted in nearly identical levels of stalled forks between the two different control samples and also for the Pol kappa-depleted samples (new Figure 1C). This demonstrates that HU-induced nucleotide deprivation is likely the major cause of stalled forks in the fork restart assay. In addition, we now show via Western blot that only by adding deoxynucleosides, but not ribonucleosides, can we down-regulate checkpoint signaling (as measured by phospho-Chk1 and phospho-RPA32) in the presence of HU to levels similar to washing off HU during the recovery step (new Figure 1D). We think this is a very nice control showing that only deoxynucleoside supplementation (not ribonucleoside) will enable recovery from checkpoint activation and elevated stalled forks in HU-treated cells. This isn’t surprising since HU is known to potently inhibit ribonucleotide reductase (RNR) to prevent the generation of deoxyribonucleotides from ribonucleotides. Adding excess ribonucleosides to the media would not circumvent this problem when RNR is already inhibited by HU.

To address the potential universal stalled fork role of Pol kappa in fork rescue, we tested whether Pol kappa is involved in fork restart using different stalling conditions, such as aphidicolin, which inhibits the DNA replication machinery without affecting the nucleotide pool level. We provide new data showing that Pol kappa does not play a role in replication restart after aphidicolin treatment (new Figure 1E). Interestingly, we found that the levels of stalled forks are higher in control samples after aphidicolin treatment compared to high dose HU (new Figure 1E), implying that recovering from stalling by direct replicative DNA polymerase inhibition is a distinct mechanism from disruption of nucleotide pool levels. We also attempted to use gemcitabine as a drug that can also inhibit ribonucleotide reductase but with a mechanism distinct from HU, and has been used as a fork stalling agent (Mini et al., 2003, Annals of Oncology; Schlacher et al., 2012). Performing the DNA fiber assay with 1μM gemcitabine yielded no CldU tracts following drug wash off (new Figure 1—figure supplement 2A), implying an inability for forks to restart, which is likely due to the difficulty of reversing the effects of gemcitabine treatment, especially when it acts as a nucleoside poison. In support of this finding, we show that washing off gemcitabine is unable to reduce phospho-Chk1 or phospho-RPA32 signaling compared with HU or aphidicolin, even when used at doses as low as 100nM (new Figure 1F and Figure 1—figure supplement 2B). This is likely due to gemcitabine’s other roles as a DNA polymerase inhibitor and chain terminator.

To address if any effects can be attributed to replication-transcription conflicts that may occur during the high-dose HU treatments, we utilized inhibitors of RNA Pol II transcription elongation (DRB and flavopiridol) in coordination with 2mM HU. Based on Cimprich and colleagues’ recent work and others (Hamperl et al., 2017, Cell; Sanz et al., 2016, Mol Cell), transcriptional R loops that lead to DNA breaks occur at transcribed genes, which could be mitigated by blocking RNA Pol II transcription elongation. As shown by western, treatment with DRB or flavopiridol (FVP), two inhibitors of CDK9, does not affect checkpoint signaling in response to HU treatment (new Figure 4—figure supplement 1B). In addition, treatment with DRB shows no change in the length of CldU incorporation under HU in both control and Pol kappa-depleted samples (new Figure 4—figure supplement 1A), showing that the slow DNA synthesis and travel of the fork under high-dose HU is not affected by replication-transcription conflicts that can be mitigated, in part, by DRB treatment.

2) When discussing the results of Figure 4A, the authors state that "PolK depletion further reduced DNA synthesis (or fork speed) during HU treatment, as measured by reduced CldU tract length adjacent to IdU tracts". Could this reduction be due to the fact that PolK depleted cells have a defect in fork restart? During the time of HU treatment, replication forks will continuously stall and restart, and the reduced efficiency of restart associated with the loss of PolK could lead to shorter CldU tracts.

This is an interesting point made by the reviewers, but if this hypothesis were correct, then one would also expect that depletion of FANCD2, which has a defect in fork restart, to also show reduced CldU tracts adjacent to IdU tracts. Depletion of FANCD2, however, increases the CldU synthesis under high-dose HU, seen in this report and others (Chen et al., 2015; Lossaint et al., 2013). It is hard to imagine that DNA synthesis during HU would be discontinuous at the leading strand with fits of stops and restarts, but we can’t formally rule out this possibility without further in vitro analysis, which we think is beyond the scope of this study. We have modified our discussion of the Figure 4A results according to the suggestion of the reviewers (subsection “PolK protects forks against SMARCAL1- and MRE11-dependent nascent DNA degradation”).

3) Figure 5. The authors should test how the cell cycle detects and the 53BP1 nuclear body formation accumulation observed upon PolK loss are affected by co-depletion of FANCD2 and PolK to confirm that these proteins act within the same pathway.

This is a critical point brought up by the reviewers. We now provide new evidence to show that co-depletion of Pol kappa and FANCD2 leads to similar levels of 53BP1 nuclear bodies as well as senescence-associated b-galactosidase staining (new Figure 6C and Figure 6—figure supplement 1B), indicating that these two proteins are acting in the same pathway after transient nucleotide deprivation to prevent genomic instability and cell cycle defects.

4) The authors show (Figure 2) that Pol kappa-deficiency does not affect Chk1 signaling in response to high dose HU treatment. This result directly contradicts a previous study (Bétous et al., 2013) and impacts ongoing discussions regarding reproducibility in science. This discrepancy is important for the field to reconcile. The authors should provide more information regarding possible differences in cell lines (genetic differences), doses, antibodies used, or other possible experimental reasons underlying differences.

We appreciate the concerns of the reviewers regarding reproducibility in science, but we have a fundamental disagreement with this policy. It shouldn’t be up to us to go out of our way to try to prove or disprove other fellow scientists in the field. The only thing we can control is our own set of results and data analysis. We can only speculate why there may be discrepancies in science, some could involve more tabloid-like discussions. But in this case, the best thing to do is to take the long view of research and science. In general, time is the greatest arbiter of science. If there haven’t been any follow-up studies regarding the role of Pol kappa and Chk1 signaling (and there hasn’t been since 2013), then most likely the effect is either too modest to be followed up by other researchers in the crowded DNA damage response field, or the effect is too difficult to capture reproducibly. Logically, if the loss of Pol kappa truly had a checkpoint defect, the Pol kappa-deficient mice would have a more severe phenotype, bordering an ATM or ATR-like hypomorphic condition. The Pol kappa KO mice have relatively mild phenotype (no developmental defects or cancer predisposition). That being said, we are currently using Pol kappa KO mice to interrogate endogenous replication stress inducers, looking at cell type-specific effects in a more endogenous setting. We hope to report this phenotype soon, but this is currently beyond the scope of our study in this manuscript.

In regards to the Bétous et al. (2013) study, there are similarities in the cell lines used and dosages of HU performed between our current study and theirs. Both show U2OS and 293T cells, and both use 2mM HU for similar time points (3 hrs in Bétous et al., versus 4 hours in our manuscript). However, there are some issues in interpreting some figures. For instance, in their manuscript regarding Figure 1C, Figure 5B, Supplementary Figure 1D, and Supplementary Figure 1F, they all show reduced total Chk1 protein levels, which could explain the decrease in the phospho-Chk1 they observed. In their Figure 1E, they attempt to show that reduced Pol kappa activity leads to reduction in phospho-Chk1 levels by expressing a catalytic dead Pol kappa in MRC5 cells. In their figure there were no untreated control to compare phospho-Chk1 levels, and the levels of kappa expression were not shown. In contrast, we have evidence that overexpression of a catalytic dead (CD) version of Pol kappa actually leads to increased phospho-Chk1 levels in untreated cells, but shows no changes of checkpoint signaling upon 2mM HU treatment (see Author response image 1). Our interpretation here is that prolonged Pol kappa usage in unperturbed cells (due to USP1 knockdown or p21 knockdown, which prevents the recycling of Pol kappa) is a bad thing, as this would lead to more fork stalling and slower replication forks, under-replicated DNA, and genomic instability (see Jones et al., 2012; Mansilla et al., 2016). Therefore, expression of a Pol kappa catalytic mutant would further exacerbate this problem, resulting in recruitment of a dead DNA polymerase to a replication fork, leading to increased fork stalling and elevated levels of checkpoint signaling. However, we feel that the effect of expressing the Pol kappa catalytic mutant in unperturbed cells is not relevant in our current study since we are primarily studying how HU-induced stalled forks are rescued by Pol kappa; therefore, we decided to leave it out of the current manuscript.

**Author response image 1. respfig1:** Showing expression of WT vs Catalytic-dead (CD) Pol kappa in Pol kappa sgRNA KO cells. Even in the absence of HU (2mM), the CD mutant has elevated Chk1 and RPA32 phosphorylation. This is in contrast to the Bétous et al. (2013) study showing that Pol kappa is required for Chk1 activation upon HU treatment.

5) Why does the expression of PolK UBZ mutant in the complementation experiment decrease the CldU track length as compared to vector (Figure 3B)? Is this statistically significant, and if so, what this may indicate mechanistically? It would be wise to add this in the Discussion section.

After re-inspecting the results for Figure 3B, we recognized an error that images from one of the slides used from the UBZ dataset were acquired at a higher resolution from the microscope, however the scale conversion to μM used was that of the standard resolution similar to the rest of the data. This has been corrected and the data updated (new Figure 4B). We sincerely apologize for this oversight. Upon the re-analysis of this data, the average track length of CldU for the Pol kappa UBZ-expressed cells is only marginally shorter than the vector control (compare avg of 10.92 kb for vector control to 9.75 kb of the UBZ mutant) with a p value of 0.03. It is possible that expression of the UBZ mutant is acting as a mild dominant negative and binding to a protein required for the protection of stalled forks. One possibility is that since the UBZ mutant still retains minimal PCNA binding activity through its PIP box, it’s possible that binding to PCNA, independent of its ubiquitin-binding capacity, could alter the PCNA dynamics at stalled forks, which somehow leads to increased nascent DNA degradation. We have modified the text to speculate on this point.

6) Figure 4D. Why FANCD2 depletion does not affect EdU foci counts while it increases the CldU tract lengths in Figure 4A?

Unfortunately, we don’t have a good explanation for this (we provide more discussion on the interpretation of our data in the subsection “PolK-mediated DNA synthesis during conditions of nucleotide starvation”). A prediction would be that with FANCD2 depletion, we would see an increase in the amount of EdU counts per foci, as there are longer CldU tracts as measured with DNA fiber analysis under HU treatment (compare Figure 5A and 5D). It is not clear why we do not observe this. We would have predicted, as pointed out by the reviewers, that the amount of EdU foci counts per nuclei correlates with the number of ongoing replication fork events in the nucleus, whereas the amount of EdU counts (molecules) per foci would correlate with the amount of EdU incorporation (DNA synthesis) at specific clusters of replication forks per event. This implies that single-molecule fork events quantified by nucleotide analog signal measurements using DNA fibers versus EdU signals via super-resolution is not based on a one- to-one direct correlation of a single elongating fork, but likely a cluster of fork events within an EdU focus as defined by super resolution. It is possible that the increased synthesis at each replication fork is ‘diluted’ by what is occurring at multiple forks in the nucleus, and so within each individual EdU focus the increase is not as significantly different. FANCD2 appears to be playing a role in supporting the number of forks undergoing active DNA synthesis as co-depletion of Pol kappa and FANCD2 reverses the number of EdU foci counts per nuclei, but maintains similar amounts of EdU counts per foci as with Pol kappa depletion alone. While Pol kappa and FANCD2 are epistatic for fork restart, other roles of FANCD2 cannot be ruled out, as it is known to interact with other HDR factors for DNA repair, and its role in inhibiting replication origin firing (Chen et al., 2015) also cannot be ruled out.